# Technical Condition Assessment of Light-Alloy Wheel Rims Based on Acoustic Parameter Analysis Using a Neural Network

**DOI:** 10.3390/s25144473

**Published:** 2025-07-18

**Authors:** Arkadiusz Rychlik

**Affiliations:** Department of Vehicles and Machinery, Faculty of Technical Sciences, University of Warmia and Mazury, Oczapowskiego 11, 10-719 Olsztyn, Poland; rychter@uwm.edu.pl

**Keywords:** wheel rims, acoustic diagnostics, neural networks, *T*
_60_, modal analysis, binary classification, deep learning, real-life AI, fatigue defect detection

## Abstract

Light alloy wheel rims, despite their widespread use, remain vulnerable to fatigue-related defects and mechanical damage. This study presents a method for assessing their technical condition based on acoustic parameter analysis and classification using a deep neural network. Diagnostic data were collected using a custom-developed ADF (Acoustic Diagnostic Features) system, incorporating the reverberation time (*T*_60_), sound absorption coefficient (α), and acoustic energy (*E*). These parameters were measured during laboratory fatigue testing on a Wheel Resistance Test Rig (WRTR) and from used rims obtained under real-world operating conditions. The neural network was trained on WRTR data and subsequently employed to classify field samples as either “serviceable” or “unserviceable”. Results confirmed the high effectiveness of the proposed method, including its robustness in detecting borderline cases, as demonstrated in a case study involving a mechanically damaged rim. The developed approach offers potential support for diagnostic decision-making in workshop settings and may, in the future, serve as a foundation for sensor-based real-time rim condition monitoring.

## 1. Introduction

Ensuring a high level of operational safety in motor vehicles requires effective diagnostic methods for assessing the condition of structural components. Among these, wheel rims—key elements of the vehicle’s suspension system—are subjected to variable and often extreme mechanical loads, including both fatigue and impact stresses resulting from dynamic interactions between the road surface and the wheel [1,2]. Light-alloy rims (typically made from aluminum or its composites) are particularly prone to fatigue crack initiation, rendering them highly susceptible to structural degradation [3].

Although the terms wheel and rim are often used interchangeably, they refer to different components: technically, the rim is the outer, narrow shell that supports the tire, while the wheel refers to the complete assembly, including the mounted tire. The tire is radially positioned by two beads and makes contact with the edge of the rim, exerting axial pressure on two flanges. These flanges are classified according to their geometric profiles.

The connection between the rim and the hub—implemented either through a solid disc or a system of spokes—further classifies rims into disc-type and spoked-type categories. A detailed illustration of a spoked rim is presented in Figure 1.

Traditional methods for assessing the condition of wheel rims rely on visual inspection and geometric measurements (e.g., radial and axial runout), which—although relatively simple—do not enable early detection of fatigue-related or structural defects [4]. Moreover, accurate evaluation typically requires tire removal, increasing both the time and cost of diagnostics. Consequently, there is growing interest in the application of acoustic methods, including impulse response analysis and time–frequency techniques, as non-invasive tools that are amenable to automation [5].

Borecki et al. [6] proposed a method for evaluating the technical condition of steel wheel rims based on vibration analysis and the use of neural networks. The recorded vibration signals were subjected to FFT transformation and feature extraction and then classified using a Self-Organizing Map (SOM) and a Multi-Layer Perceptron (MLP). These networks enabled the differentiation of three technical condition classes: new, used and serviceable, and unserviceable. The authors demonstrated that this approach can be effectively applied in workshop conditions without the need for invasive diagnostic methods.

Although the vibroacoustic approach using conventional vibration sensors and neural networks [6] has shown high effectiveness in service conditions, it requires direct contact with the tested object and specific measurement conditions. A less invasive and remotely applicable alternative may lie in the analysis of acoustic parameters such as reverberation time (*T*_60_), sound absorption coefficient (α), and acoustic energy (*Ε*). These parameters reflect the propagation properties of acoustic waves within the tested structure and can serve as sensitive indicators of changes in stiffness and structural integrity of the rim [7,8]. Previous studies have shown that these features are sensitive to both the type and location of damage, making them a promising basis for the development of condition classification models [9,10,11].

In response to the growing need for automation in diagnostics and the elimination of subjective factors, artificial intelligence (AI) tools—particularly neural networks—are increasingly being employed. Deep learning (DL)-based models enable effective classification of high-dimensional measurement data and exhibit robustness to noise and disturbances [7,12]. Applications of convolutional neural networks (CNNs) and Long Short-Term Memory (LSTM) architectures have demonstrated their effectiveness in the analysis of unstructured signals and modal feature extraction [13,14].

The method developed in this study is based on an acoustic analysis of wheel rims both obtained from operational use and subjected to controlled wear under laboratory conditions. Three diagnostic features are extracted from the appropriately processed acoustic signals: the reverberation time (*T*_60_), sound absorption coefficient (α), and acoustic energy (*E*). The normalized and standardized input data are then classified using a neural network trained on laboratory data. The classification assigns each sample to one of two classes—“serviceable” or “unserviceable”—based on the logit (z), the classification probability, and the distance from the decision boundary [15,16]. The interpretation of these values is presented below:Logit (z): The output value of the activation function in the final layer of the neural network, prior to its transformation into a probability. It is typically a real number, positive or negative, where the sign indicates the side of the classification and the magnitude reflects the confidence level.Classification probability: The result of applying the logistic (sigmoid) function to the logit, ranging between 0 and 1, and interpreted as the likelihood that the rim belongs to the “unserviceable” class.Distance from the decision boundary: The absolute difference between the classification probability and the decision threshold of 0.5 (i.e., |P − 0.5|); the greater the distance, the higher the confidence in the classification, while values close to zero indicate that the sample lies near the boundary between the two classes.

The objectives of this study are as follows:
To validate the usefulness of acoustic parameters as indicators of the technical condition of light-alloy wheel rims;To evaluate the effectiveness of classification using a neural network trained on laboratory data;To test the robustness of the method against borderline cases (i.e., mechanically damaged rims not present in the training dataset);To present a diagnostic application enabling classification of real-world rim samples obtained from the market.

The structure of this article is as follows: Section 2 describes the materials and methodology, including the WRTR wear-testing stand, the ADF diagnostic system, the FEM model implemented in the ElmerGUI environment, and the architecture of the neural network. Section 3 presents the results and their discussion, including classification of field-acquired data. Section 4 outlines the conclusions and directions for further development of the proposed diagnostic methodology.

## 2. Materials and Methods

### 2.1. Materials

This study focused on two groups of automotive wheel rims made of light alloys.

The first group consisted of four factory-new OEM-category rims, each representing a different size. The second group comprised rims that had undergone regular use on passenger vehicles. These used rims varied in size, mileage, part category (OEM, Q, P), technical condition, and age.

The used rims were collected during seasonal tire replacement at a tire service facility over two consecutive seasons (summer and winter), allowing for the acquisition of a sample set with varying degrees of wear and technical condition.

Table 1 presents the general characteristics of the tested wheel rims.

### 2.2. Wheel Resistance Test Rig (WRTR)

The Wheel Resistance Test Rig (WRTR) method is used to induce controlled degradation of wheel rims. The test stand was designed and constructed in accordance with the requirements of the ECE R124 regulation [17]. In this study, the WRTR was used to degrade the new rims described in Section 2.1.

A general schematic of the WRTR test stand for evaluating the bending resistance of a wheel rim under torque is shown in Figure 2. The stand consisted of a supporting frame (2), to which the tested rim (1) was mounted. A shaft (3) was terminated at one end with a hub, onto which the disc of the tested rim was fixed. The opposite end of the shaft was connected to an unbalanced mass mechanism (4) driven by an electric motor. The imbalance force it generated, acting along the shaft axis (3), produced a bending moment applied to the tested rim.

To measure the amplitude of shaft oscillations, an eddy current sensor (5) was used in conjunction with a data acquisition unit (6).

The number of load cycles and operation of the test stand were controlled by a control module (7).

To monitor the oscillations of the load shaft in the WRTR setup, a ZPW Vibration Meter was used. According to the manufacturer, the measurement error did not exceed 2% of the recorded value. For the maximum oscillation amplitudes observed during testing, this corresponded to a measurement uncertainty of approximately ±0.04 mm.

During the bending-under-torque test using the WRTR setup, lateral forces acting on a vehicle’s wheel during cornering were replicated. The wheel rim was rigidly mounted to the test stand frame, while a bending moment was applied to the disc mounting area via the hub—mimicking real operational conditions of the vehicle for which the rim was intended.

The load level applied to the rims in the WRTR stand was set at approximately 50% of the bending moment required by regulation [17], with values adjusted for the specific rim size, application, and material type.

For example, for a 15-inch aluminum alloy rim, the applied moment was approximately 1420 Nm (i.e., 50% of the reference value) at a forcing frequency of 3.06 Hz.

The minimum number of load cycles was defined in accordance with [17] and set at 1.8 × 10^6^. Within the examined load range, the occurrence of structural cracking in the rim was considered unacceptable.

The torque applied to the hub bolts was set in accordance with the rim manufacturer’s specifications and amounted to 120 Nm. This value was checked upon both tightening and loosening using a calibrated torque wrench.

The criteria for evaluating the rim’s technical condition during the bending resistance test followed the requirements of regulation [17]. Unlike homologation testing, where the test ends upon reaching a predefined number of cycles, in this study the test was terminated when a structural discontinuity (crack) appeared in the rim.

Each tested rim was mounted to the hub of the test rig using a new set of mounting bolts.

The duration of a single test was defined as one testing cycle, typically lasting about 12 or 24 h depending on the test parameters.

After each test cycle, the rim was removed from the rig and subjected to a visual inspection—without the use of measuring instruments.

Subsequently, the acoustic and geometric parameters of the tested rim were identified.

#### Wear Procedure for New Wheel Rims

The wear procedure for wheel rims is presented in Algorithm 1.
**Algorithm 1.** Cyclic loading procedure for wheel rims on the WRTR test stand.1: Prepare the WRTR test stand for operation2: Mount the test rim to the hub using a new set of bolts (tightening torque: 120 Nm)3: Configure test parameters:     – Set the number of loading cycles     – Select cycle duration: 12 or 24 h     – Define the bending moment level (50% of the value specified in [17])4: Start the test stand motor and initiate bending moment generation via the unbalanced mass5: Monitor the test operation:     – Count the completed loading cycles     – Record shaft oscillations using the eddy current sensor6: Upon cycle completion, stop the motor and disconnect power7: Verify the torque on mounting bolts (control at 120 Nm), then dismount the rim8: Perform visual inspection of the rim (without measuring instruments)9: Conduct acoustic and geometric measurements of the rim using the ADF system10: Check for structural cracking of the rim:     – If cracking is present: terminate the procedure     – If not: return to step 2 and begin a new cycle

### 2.3. Acoustic Diagnostic Features (ADF) Method

The author-developed Acoustic Diagnostic Features (ADF) method is used to identify the acoustic and geometric parameters of wheel rims. The ADF setup was employed to extract diagnostic features from both new rims and rims degraded using the WRTR stand. The method was also applied to rims worn through regular service life.

A general view of the ADF diagnostic stand is shown in Figure 3.

The core of the technical condition assessment of the wheel rim involved analyzing the acoustic signal recorded by the RG-50 microphone (ROGA Instruments, Nentershausen, Niemcy) (3), positioned perpendicularly to the axis of the shaft and self-centering hub fixture (2). The measurement process began with mounting the tested wheel rim (1) in the fixture. The vertical orientation of the shaft and the use of a self-centering fixture ensured gravitational clamping of the rim disc against the hub, as well as concentric (axial) alignment of the rim relative to the measurement system.

After the rim was mounted in the ADF stand’s self-centering fixture, a gravitational excitation force (impact) was applied to the rim using a hammer (4)—see Figure 3. The impact was delivered to the lower edge (flange) of the rim in a direction tangential to the plane of the wheel disc. The direction of vibration excitation remained perpendicular to the orientation of the microphone, allowing for the recording of the rim’s acoustic response in the axial direction of propagation.

As a result of the impact, amplitude–time waveforms of the acoustic pressure generated by the rim’s resonant sound were recorded.

During the excitation and measurement process, the wheel rim remained stationary (i.e., it did not rotate).

The measurement of rim bead runout was carried out in the next step using the same fixture and a Keyence LV-7080 non-contact displacement sensor (5)—see Figure 3.

During this procedure, distances were recorded in two planes, axial (A) and radial (R) for both the inner (IN) and outer (OUT) bead seats of the rim, relative to the centering bore. The acquired data were then correlated with the angular position of the rim.

The acoustic measurement system of the stand was equipped with a ROGA RG-50 free-field microphone, with a frequency response of 20 Hz to 4 kHz (±1.0 dB) and 4–20 kHz (±1.5 dB). The sampling rate used in the experiments was 8 kHz, which enabled the analysis of signals up to 4 kHz—fully within the microphone’s most accurate measurement range (±1.0 dB).

This ensured that the recorded data fell within the most precise operating range of the measurement system. At the same time, the selected sampling frequency significantly exceeded the minimum requirement defined by the Shannon theorem (fs > 2fmax), where the maximum identified vibration frequency of the rim was approximately 500 Hz—meaning that a theoretical minimum of 1 kHz sampling would have sufficed.

The use of 131,072 samples enabled high time–frequency resolution. The frequency resolution was 0.061 Hz. The time resolution, determined by the 8 kHz sampling rate, was 0.000125 s (125 µs), meaning that one sample was recorded every 125 microseconds. These parameters allowed for detailed analysis of signals in both the time and frequency domains.

The RG-50 microphone was interfaced with an NI USB-6210 data acquisition card (National Instruments Corp., Austin, TX, USA). The measurement chain was calibrated using an acoustic calibrator set to a reference level of 94 dB SPL at 1 kHz prior to each measurement series. Calibration was carried out according to the microphone manufacturer’s guidelines and the measurement system requirements.

The excitation energy for rim vibration was delivered using a gravitational impactor with an impact energy of 1 ± 0.1 J (mass = 0.425 kg, drop height = 0.240 m) generated from an initial hammer deflection of 45°.

Axial and radial runout measurements were performed using a Keyence LV-7080 laser displacement sensor, operated via dedicated LJ-Navigator2 (Version 2.02.00, Keyence Corporation, Osaka, Japan) software. According to the technical documentation, the repeatability of Z-axis (height) measurements is 0.5 µm, and the linearity is ±0.1% of full scale (F.S.). The measurement accuracy for the rim bead’s geometric parameters—taking into account electronic readout and measurement uncertainty—was ±0.01 mm [18].

All acquired data were visualized, processed, analyzed, and archived using the MATLAB environment R2022b (The MathWorks Inc., Natick, MA, USA). An in-house analytical procedure was developed to extract key features of the acoustic signal from the tested rim component, including the following:*T*_60_—reverberation time of the rim, defined as the time required for the signal to decay by 60 dB;Alpha (α)—acoustic wave absorption coefficient;Acoustic energy (*E*)—defined as the square of the root mean square (RMS) amplitude of vibrations;Maximum axial (*A*) and radial (*R*) runout of the rim bead, measured relative to the centering bore, for both the inner (IN) and outer (OUT) sides of the rim.

#### Test Procedure Using the ADF System

After each test cycle, the wheel rim was dismounted from the WRTR stand and subjected to visual inspection. In the case of rims obtained from in-service vehicles, the tireless rim underwent visual evaluation including the rim body, disc, and hub areas. If mechanical cracks, signs of repair, or straightening were detected, the rim was excluded from the study group.

After positive verification of the rim’s general condition, it was mounted on the ADF test stand and subjected to the measurement procedure.

The measurement and acoustic signal analysis procedure is presented in Algorithm 2.

Each measurement was repeated five times, with the tested rim being dismounted and remounted on the ADF stand shaft before each repetition. After completing the measurements, the rim was remounted on the WRTR stand and subjected to subsequent loading cycles. These steps were repeated cyclically until the rim test process was concluded, i.e., when structural damage (cracking) of the tested rim occurred during WRTR testing.

In the case of in-service rims, after measurements were completed, the rim was reassembled with a tire, balanced, and mounted back on the vehicle.
**Algorithm 2.** Procedure for acoustic feature extraction for wheel rim diagnostics1: Initialize the ADF measurement station2: Configure measurement devices:     – Set sampling frequency of NI USB-6210 to 8 kHz     – Set initial hammer angle to 45°, corresponding to 1 ± 0.1 J impact energy3: Mount the test wheel rim in the self-centering fixture4: Ensure static positioning: gravitational contact, no rotation5: Apply mechanical excitation by striking the rim’s edge tangentially6: Record acoustic response from ROGA RG-50 microphone for 4 s7: Store signal data s(t), t ∈ [0, 2]8: Use Keyence LV-7080 sensor to measure radial and axial runout:     – Record distance in axial (*A*) and radial (*R*) directions     – Measure for both IN and OUT sides relative to the centering hole9: Calculate RMS value of s(t) over 2 s10: Compute *T*_60_ by estimating the 60 dB decay time from s(t)11: Calculate absorption coefficient α using Sabine’s formula12: Compute acoustic energy *E* = RMS^2^13: Store results in structured data table:     – [*T*_60_, α, *E*, *A*_IN_, *A*_OUT_, *R*_IN_, *R*_OUT_]14: Save table to MATLAB Workspace for further analysis

### 2.4. Numerical Modeling in Elmer

The Elmer software package, a suite of solvers for various differential equations, was used for the simulation studies. Details on the solvers and assumptions used in the analysis are provided later in the text.

To analyze the coupled mechanical–acoustic phenomena occurring in the system composed of the wheel rim and the enclosed air volume, a finite element method (FEM)-based approach was employed, using the open-source ElmerGUI package (compiled on 10 February 2025; CSC–IT Center for Science Ltd., Espoo, Finland) [18].

Four types of solvers available in Elmer were utilized in the numerical analyses, each corresponding to different coupling mechanisms and types of dynamic analysis:Linear Elasticity + EigenSolve: Used to solve the modal problem in the solid domain (rim). This solver addresses the linear equations of motion for an elastic body in the form
(1)$$\rho \frac{\partial^{2}u}{\partial t^{2}}=\nabla \cdot \sigma +f,$$
where *ρ* is the material density, **u** is the displacement vector, *σ* is the stress tensor, and **f** is the body force vector.

The EigenSolve solver transforms the above equations into an eigenvalue problem, enabling the identification of natural vibration frequencies and their corresponding mode shapes.

Acoustics + EigenSolve: Modal analysis in the acoustic domain, based on the solution of the Helmholtz equation:
(2)$$\nabla^{2}p+\left(\frac{\omega^{2}}{c^{2}}\right)p=0,$$
where *p* is the acoustic pressure, *ω* is the angular frequency, and *c* is the speed of sound.

This solution allows the determination of the resonant frequencies of the enclosed air volume (inside the rim) and the associated spatial distributions of the pressure field.

Linear Elasticity + Acoustics + EigenSolve: Coupled structural–acoustic modal analysis, also referred to as Acoustic–Structure Interaction (ASI). This approach accounts for energy transfer from the solid domain (rim) to the acoustic medium (air) through a shared contact surface.

The simulation incorporates continuity conditions for pressure and velocity at the fluid–structure interface. A detailed description and application of the corresponding solvers can be found in [18,19].

Transient Elasticity + Transient Acoustics: Time-domain dynamic analysis (nonlinear), used to simulate the system’s response to an impulse excitation. This involves solving the equations of motion and the acoustic wave equation for a time-varying input:
(3)$$\rho \frac{\partial^{2}u}{\partial t^{2}}=\nabla \cdot \sigma +f\left(t\right),\;\;\frac{1}{c^{2}}\frac{\partial^{2}p}{\partial t^{2}}=\nabla^{2}p+S(t),$$
where **f**(t) is the time-dependent body force and *S*(t) is the excitation source (e.g., hammer impulse).

This approach enables the analysis of the temporal evolution of parameters such as acoustic pressure, reverberation time *T*_60_, and acoustic energy in response to a known excitation signal.

Due to the complexity of the subject, a detailed presentation of the governing equations and their numerical solutions is beyond the scope of this paper. The reader is referred to the Elmer software documentation [18] for further details.

The geometric model reflects the actual shape of a 4.5Jx13H2 rim made of aluminum alloy and explicitly includes a domain representing the air volume enclosed within the rim. This domain was treated as a compressible medium.

A key aspect of the modeling was the accurate representation of the structure–fluid interface, enabling the transfer of mechanical energy from the rim to the acoustic medium in the form of pressure waves.

The simulations employed both modal analysis, to determine the natural frequencies of the coupled system, and time-domain analysis, to simulate its response to impulsive excitation.

To enhance the realism of the model, viscoelastic damping was introduced in the material of the rim using Rayleigh damping coefficients [20].

Additionally, acoustic wave reflections at the boundaries of the air domain were considered by treating these boundaries as rigid surfaces, in accordance with Neumann boundary conditions (i.e., acoustically hard walls) [21].

The objective of the analysis was to identify acoustic parameters such as reverberation time (T_60_), sound absorption coefficient (α), and acoustic energy (E) as functions of time and spatial distribution. To achieve this, impulse response simulations were performed at selected points of the model corresponding to the positions of measurement microphones placed along the x, y, and z axes.

Based on the time-domain waveforms of the acoustic pressure recorded at these locations, local reverberation times were estimated using the Schroeder integration method, the sound absorption coefficient was calculated using Sabine’s formula, and the spatial distribution of acoustic energy within the domain was determined.

The aim of this numerical study was to assess whether changes in the technical condition of the rim—such as cracking of the rim body or spokes or geometric asymmetry—affect the acoustic parameters, namely, *T*_60_, sound absorption coefficient (α) and energy distribution (*E*).

A physical model of the 4.5Jx13H2 rim was digitized using a SMARTTECH MICRON3D optical 3D scanner (SMARTTECH Sp. z o.o., Warsaw, Poland). The obtained mesh model was postprocessed (cleaned) and exported in a format compatible with ElmerGUI.

Boundary conditions reflecting the gravitational, non-rigid mounting of the rim were applied in the simulation, mimicking the actual conditions of acoustic testing on the ADF station.

To identify modal parameter changes induced by different geometric and structural conditions, the nominal digital model of the rim was modified to generate specific variants incorporating selected structural damage or geometric deformations.

Table 2 summarizes the characteristics of the analyzed cases, including both structural defects (e.g., cracks) and geometric asymmetries introduced into the digital model of the 4.5Jx13H2 rim.

#### 2.4.1. Numerical Model in ElmerGUI

Table 3 summarizes the key parameters of the numerical model and the simulation settings applied in the ElmerGUI environment. Material properties, mesh characteristics, dynamic analysis time parameters, excitation method, and the locations of acoustic pressure sampling points were included.

The force impulse amplitude was estimated based on an impact energy of 1.0 J and an impulse duration of 1 ms. A dynamic contact condition was assumed, producing local displacements in the range of 1–2 mm.

Due to limitations in data handling by the ElmerGUI environment, meshes with 1 mm element size could not be correctly imported into the simulator. Therefore, a coarser mesh with 5 mm element size was adopted for the analysis, which allowed successful geometry loading and completion of the computations.

A mesh convergence test confirmed that refining the mesh from 7.5 mm to 5 mm did not significantly affect the analysis results. Differences in the calculated natural frequencies and acoustic parameters (*T*_60_, α) remained within 5%, confirming that a 5.01 mm mesh provides sufficient accuracy while reducing computational effort.

In the modal analysis of the rim–air system, the first 10 eigenmodes were computed, corresponding to frequencies up to approximately 5 kHz. This frequency range includes the most significant structural and acoustic vibrations affecting the *T*_60_ and Alpha (α) parameters.

#### 2.4.2. Analysis Procedure in ElmerGUI

A summary of the applied analysis methods, solver configurations in the ElmerGUI environment, and acoustic parameters is presented in Table 4. The table outlines the computational approaches used, the configuration of Elmer solvers, and specific settings adopted for the analysis of mechanical and acoustic phenomena. Additionally, typical output data are indicated for each processing stage.

The numerical model employed a single acoustic pressure measurement point, corresponding approximately to the microphone position used in experimental tests on the ADF test stand. This point was located at the geometric center of the inner volume of the wheel rim.

The described numerical model provides the basis for further analysis of how the technical condition of the wheel rim affects selected acoustic parameters. The following sections present the results of modal and time-domain simulations, as well as comparisons between reference and defect-induced cases.

### 2.5. Identification of Features in Field-Used Rims

Acoustic and geometric parameter measurements of in-service wheel rims were carried out using rims obtained during seasonal tire changes (from summer to winter tires or vice versa) at a service facility. The rims were dismounted from vehicles and, after tire removal, subjected to technical inspection and further measurements.

Before measurements on the ADF test stand, each rim underwent a preliminary visual inspection for cracks, scratches, impact marks, signs of straightening, or any mechanical repairs. Additionally, axial and radial runout measurements were performed using non-contact sensors. Rims exhibiting runout values exceeding 1.5 mm (on either the inner or outer side) were excluded from further analysis. This threshold was selected based on workshop practice and technical guidelines (e.g., manufacturer recommendations [22] and industry standards [23]), which state that acceptable rim runout should not exceed 1.25 mm.

Rims that met the allowable runout criteria but showed visible mechanical damage (e.g., cracks, repair marks, straightening) were also disqualified from further testing.

Only rims that exhibited no mechanical damage and fell within the allowable geometric deviation range were classified as undamaged. All measurements (*T*_60_, α, acoustic energy (*E*), geometric parameters) were performed in accordance with the measurement procedure described in Algorithm 2.

Table 5 presents the classification of wheel rims qualified for or excluded from this study. Depending on geometric condition and presence of mechanical damage, rims were assigned to one of three categories: undamaged (passed), damaged (failed), or rejected.

### 2.6. Neural Network-Based Classification Method

To enable the automatic assessment of the technical condition of wheel rims, a classification approach based on an artificial neural network (ANN) was employed. The task of the network was to determine the usability of a rim (pass/fail) based on an acoustic feature vector. The training dataset was constructed using data obtained from laboratory experiments (WRTR test rig and ADF diagnostic station) as well as measurements of rims used under real operating conditions.

Each examined rim was represented by a three-dimensional feature vector, $X=\left[T_{60},\alpha ,E\right]$, where *T*_60_ is the reverberation time, α is the sound absorption coefficient, and *E* is the acoustic energy.

Geometric parameters were deliberately excluded from the feature vector, as they are indirectly related to the dynamic–acoustic features above. The aim of the method was to develop a diagnostic approach based solely on non-contact, acoustically obtainable data.

The training dataset consisted of rims subjected to controlled degradation in laboratory settings—from their initial, new condition to the onset of damage. Each instance was labeled as either pass (label 1) or fail (label 0) based on visual inspection of the rim’s condition. This process resulted in a labeled dataset suitable for supervised learning of the neural network.

Next, the structure of the network was defined, training was conducted, and a diagnostic application was developed for real-time classification of new cases using the trained model.

The classification model was implemented as a deep neural network built and trained in MATLAB. The network accepted a three-dimensional vector of acoustic features as input, representing a single rim instance.

The input data were first subjected to normalization with respect to reference values, followed by standardization using the z-score method. This preprocessing ensured scale consistency and enhanced the effectiveness of the training process. Additionally, Principal Component Analysis (PCA) was performed to evaluate the relevance of each feature and its contribution to overall data variance.

The dataset was split into a training set (70%) and a test set (30%), maintaining the class balance across both subsets. Each data point was labeled with a binary target: “pass” (1) or “fail” (0).

A deep neural network classification model was developed and tailored to handle the three-dimensional input feature vector [*T*_60_, α, *E*]. The network architecture consisted of the following:An input layer for three features;Two fully connected hidden layers with 128 and 64 neurons, respectively, both using the ReLU activation function;An output layer with a softmax activation, enabling binary classification into either the “pass” or “fail” class.

A block diagram illustrating the architecture of the neural network is presented in Figure 4.

The training of the network was carried out using the Adam optimization algorithm, with 50 epochs and a mini-batch size of 32. The initial learning rate was set to 0.001. A validation process was applied using the test set (30% of the total data), allowing for real-time monitoring of model performance and reducing the risk of overfitting.

After training, model performance was assessed using classification accuracy and a confusion matrix. In addition, standard classification metrics such as precision, recall, and F1-score were calculated for each class. The loss function used was Binary Cross-Entropy, and the final model was saved in a .mat format, including the class centroids and label names, enabling future deployment within a real-time diagnostic system.

The architecture of the neural network was selected based on the number of input features and the size of the dataset. As each wheel rim was described by three acoustic features (*T*_60_, α, *E*), and the number of data instances was approximately 100, a moderately deep network with two hidden layers was chosen. This configuration ensures the capacity to model nonlinear relationships while limiting the risk of overfitting. The choice of training parameters—such as the use of Adam optimizer, number of epochs, and mini-batch size—was made empirically, based on training stability and validation quality

## 3. Results and Discussion

### 3.1. Experimental Plan

In this study, a comprehensive approach was adopted to assess the technical condition of automotive wheel rims made of lightweight alloys. The experimental procedure was divided into seven main stages, covering both controlled degradation of new samples and analysis of rims collected from real-world vehicle usage.

A key component of the research was the application of a custom-designed Acoustic Diagnostic Features (ADF) test bench and the ElmerGUI simulation environment for the identification of acoustic parameters. The collected data were subsequently used to train and validate a deep neural network classifier capable of distinguishing between serviceable and unserviceable rims based solely on acoustic features.

The complete structure of the experimental procedure is illustrated in Figure 5.

### 3.2. Results of Rim Fatigue Testing on the WRTR Test Stand

Table 6 presents a summary of the basic information concerning wheel rims subjected to bending fatigue tests during steering on the WRTR test stand. The table includes the rim type designation, the total number of completed loading cycles (*N*_ct_, in millions), the total test duration (*T*_t_), the number of cycles per single test (*N*_cs_), and the number of state vectors used to train the neural network—with distinction between cases classified as serviceable and unserviceable.

Table 6 summarizes four types of wheel rims subjected to fatigue tests on the WRTR test stand. Each rim was tested over multiple cycles, ranging from 20 to 29 depending on the size and wear level. The total number of measurement cycles ranged from 3.30 × 10^6^ to 7.13 × 10^6^, exceeding the minimum regulatory requirements specified in ECE R124. Based on the collected data, state vectors were generated and used to train the neural network—a total of 98 samples were obtained (83 classified as “serviceable” and 15 as “unserviceable”), providing a balanced and representative training dataset for classification algorithms.

Each acoustic measurement was repeated five times per rim under identical conditions. The resulting standard deviations of T_60_, α, and E were typically within 2–4% of the mean values, demonstrating good repeatability of the ADF system.

For the classification task, the mean values were used as input features. Measurement variability was monitored to detect and exclude outlier acquisitions. Although uncertainty propagation was not explicitly modeled in this study, its integration into future versions of the diagnostic pipeline is planned.

For detailed analysis, rim 4.5Jx13H2 was selected—the same model that was subjected to simulation tests in the ElmerGUI environment. The wear progression and the dynamics of diagnostic parameter changes for this rim were comparable to those observed in the other test cases.

In a single 24 h test cycle, this rim was subjected to approximately 264,380 load cycles. Assuming the rim was fitted with a standard tire with a rolling circumference of about 2 m, this would correspond to a curved-path length of approximately 530 km.

For reference to real-world conditions, it was assumed—in accordance with the literature [24,25] —that the ratio of curved-path driving time to straight-line driving time can be estimated based on a typical hourly driving profile: on highways, a driver performs approximately 10 lane changes per hour (each lasting ~4 s), while in urban environments, this number can exceed 100 maneuvers per hour. This means that the ratio of distance traveled in straight-line motion to that in curved motion ranges from approximately 10:1 (urban driving) to 100:1 (highway driving). This comparison highlights the severe loading conditions to which a wheel rim is subjected during the laboratory WRTR test.

Figure 6 presents a graphical interpretation of the diagnostic parameters identified for a new 4.5Jx13H2 wheel rim. These parameters—reverberation time (*T*_60_), sound absorption coefficient (α), and acoustic energy (*E*)—are shown as mean values from five measurements, with standard deviations marked. The degradation patterns observed for the remaining tested rims followed similar trends.

To improve clarity, the diagnostic parameters are plotted as a function of three variables: total number of measurement cycles (*N*_ct_), total test duration (*T*_t_ [h]), and the estimated length of the curved path traveled (in [m × 10^3^]), assuming a tire rolling circumference of 2 m. These values illustrate the impact of progressive operational loading on the acoustic condition of the rim under controlled WRTR testing conditions.

As part of the conducted research, a total of 19 complete test cycles were performed on the 4.5Jx13H2 rim at the WRTR test stand, corresponding to a total load count of Nct = 3.30 × 10^6^ cycles. The total test duration amounted to approximately 300 h, which—assuming a tire rolling circumference of 2 m—corresponds to an estimated arc-driven distance of approximately 6600 km.

The damage was confirmed after removing the rim from the WRTR stand through visual inspection of its surface. The crack appeared on the rear edge of one of the rim spokes, on both sides of the cross section (Figure 7a), originating near the profile edge. In subsequent test cycles, crack propagation extended to adjacent spokes, ultimately resulting in the rupture of a significant material section (Figure 7b).

The damage observed in the tested rim (Figure 7) was identified as a fatigue crack based on its morphology and progression, consistent with fatigue mechanisms described in previous studies [3]. This interpretation was further supported by experimental evidence from the WRTR test. A gradual increase in the oscillation amplitude of the load shaft, measured by the shaft oscillation sensor (element 5 in Figure 2), indicated progressive bending stiffness degradation. This trend, observed over time and quantified as exceeding 4% loss in stiffness at approximately 2.5 × 10^6^ cycles, is characteristic of fatigue damage accumulation in alloy rims. The combination of sensor data and failure morphology supports the conclusion that the failure mechanism was fatigue-driven.

The minimum required service life of the tested rim, in accordance with the regulation [17], is 1.8 × 10^6^ load cycles. The recorded number of cycles before crack initiation was higher, indicating a conservative design of the analyzed rim. It is important to emphasize that all tested rim specimens exceeded the minimum cycle threshold specified in the regulation [17], confirming their high fatigue resistance under the examined loading conditions.

Analyzing the trends of diagnostic parameters shown in Figure 6, clear tendencies can be observed, indicating the influence of the fatigue process on the acoustic characteristics of the 4.5Jx13H2 wheel rim.

T_60_ remained stable (3.0–3.5 s) until the 13th test cycle (i.e., ~2.5 × 10^6^ cycles). In this range, the standard deviation of T_60_ is moderate, indicating consistent results and stable dynamic properties of the rim. From the 13th cycle onward, there is a significant decrease in T_60_, reaching a minimum below 1.0 s in the final phase of testing, which clearly correlates with the onset of fatigue crack propagation (visually confirmed in Figure 7). Along with the decrease in value, there is a notable increase in measurement uncertainty—growing error bars indicate variability in the dynamic response of the rim, typical for materials undergoing degradation.

A similar trend is observed for the acoustic energy parameter E. Initially, its value fluctuates around 300–400 kPa^2^·s, with moderate standard deviation. As the damage progresses, the energy significantly decreases, reaching values below 100 kPa^2^·s in the final stage of the test. This distinct drop indicates a deteriorating ability of the rim to store and transmit acoustic energy, which may be a direct result of changes in stiffness and structural integrity of the material.

An opposite trend is observed for the absorption coefficient α. Initially low (below 0.5), its value remains relatively constant up to approximately the 13th cycle, after which a sharp increase is recorded—reaching values as high as 4.5—indicating increased acoustic energy dissipation in the damaged rim structure. The accompanying rise in standard deviation suggests an unstable energy distribution in the tested structure, a typical effect of crack propagation or the emergence of nonlinearities in the material’s dynamic response.

The observed changes confirm the high sensitivity of acoustic parameters—particularly *T*_60_ and α—to fatigue-induced damage in the wheel rim. These signals may provide an effective basis for technical diagnostics and prediction of loss of component functionality under real operating conditions.

The analysis of the results obtained for the 4.5Jx13H2 rim indicates that all analyzed diagnostic parameters—reverberation time (*T*_60_), sound absorption coefficient (α), and acoustic energy (*E*)—respond to the technical condition changes occurring during cyclic loading of the rim. In particular, a sudden change in characteristics is noticeable around 2.5 × 10^6^ cycles, coinciding with the moment when the first structural damage to the rim is identified. This behavior may suggest the existence of a critical point beyond which clear symptoms of material degradation begin to appear.

The parameter α demonstrated the highest sensitivity to changes in the technical condition of the rim, with a strong increase in value in the final phase of the test, making it potentially the most effective early-warning indicator of damage. At the same time, the increasing variability in the standard deviation of this parameter during the final test phase points to the instability of the acoustic signal accompanying the ongoing damage. The parameter E exhibited a downward trend in the final phase, while T_60_ remained relatively stable until the point of failure.

These results confirm the validity of using acoustic signals for non-invasive assessment of the technical condition of vehicle wheel rims, particularly in the context of fatigue wear. The proposed approach may serve as a method for early detection of rim unserviceability. However, it should be emphasized that the obtained results concern only a few representative rims, and full validation of the method requires further studies involving a larger number of cases and rim designs.

It is also important to note that during tests conducted on the WRTR test stand, the geometric asymmetry of the rim’s flanges was not monitored. The design of the test stand and the method of securing the rim do not reflect the actual operating conditions of a wheel in a vehicle, particularly in terms of tire–road contact and interactions resulting from geometric asymmetries. The main loads are concentrated at the structural boundary between the hub and the rim’s spokes, making this area particularly sensitive to fatigue processes.

To deepen the interpretation of the results and verify the effectiveness of the applied numerical modeling, the following subsection compares experimental findings with the results of simulations carried out in the Elmer FEM environment.

### 3.3. Technical Condition Classification of the Wheel Rim—Neural Network Trained on WRTR Data

The feature vector for classification consisted of three acoustic parameters: reverberation time (*T*_60_), absorption coefficient (α), and acoustic energy (*E*). These were selected after comparative analysis of several time- and frequency-domain features.

To evaluate the contribution of individual features to the classification process, we conducted an analysis of feature importance using logistic regression coefficients. Each feature was standardized prior to training, and the resulting absolute regression coefficients were normalized to reflect relative importance.

This approach, while simpler than SHAP, provides an interpretable ranking of the predictors. As shown in Figure 8, the absorption coefficient (α) contributed the most to the classification decision, followed by the reverberation time (*T*_60_) and acoustic energy (*E*).

This selection was further supported by prior experimental work in the WRTR framework, where these parameters showed strong relationships with fatigue-induced structural changes in aluminum alloy rims.

To automate the process of assessing the technical condition of vehicle rims, a deep learning (DL) neural network was developed and trained using data obtained from fatigue wear tests conducted on the WRTR rig, as well as acoustic feature identification from the ADF test bench. Each rim was described using a three-dimensional feature vector composed of reverberation time (*T*_60_), sound absorption coefficient (α), and acoustic energy (*E*). Based on visual inspection and observed wear processes, each rim was labeled as either “fit for use” or “unfit”.

The neural network was designed as a binary classifier with the following architecture:Input layer: 3 diagnostic features (T_60_, α, E).Two hidden layers: A fully connected layer (128) with ReLU activation followed by a fully connected layer (64) with ReLU.Output layer: A fully connected layer (2), softmax activation, and classification layer.

The training used the Adam optimizer with the following parameters:
Number of epochs: 50.Mini-batch size: 32.Initial learning rate: 0.001.Data split: 70% for training, 30% for validation.

Input data were normalized relative to reference values prior to training: T_60_ = 5 s, α = 6, E = 5 × 10^6^ Pa^2^·s. Then, the data were standardized using the z-score method. As part of the initial analysis, Principal Component Analysis (PCA) was also conducted, confirming the relevance of all three features. The first principal component explained 90.68% of the variance, the second 8.40%, and the third 0.92%, which collectively ensured complete representation of the input data structure while preserving all three features.

The training dataset consisted of 98 rim samples (classified as either “fit” or “unfit”) obtained from the WRTR laboratory fatigue tests.

The dataset was randomly split: 70% of the samples (69 instances) were used for training, 30% (29 instances) for testing the classifier’s performance.

Detailed distribution:Training set: 59 fit/10 unfit samples.Test set: 24 fit/5 unfit samples.

After training, the network achieved very high classification accuracy. The confusion matrix showed complete agreement with the reference labels (TP = 24, TN = 5, FP = 0, FN = 0). All classification metrics (precision, recall, F1-score) for both classes reached their maximum possible values (1.0). Additionally, the low Binary Cross-Entropy loss (BCE = 0.0199) indicated excellent model fit and prediction stability.

No misclassifications were identified in the test set, and the confusion matrix confirmed perfect class separability. The trained network was saved in the file trainedNet and used in the next stage for evaluating the technical condition of rims collected from field use (see Section 3.5).

The classification results obtained using the deep neural network trained on laboratory data (WRTR + ADF) confirmed the high effectiveness of the proposed method, with consistent performance across all quality metrics and the same low loss value (BCE = 0.0199), indicating no signs of overfitting.

It is also important to note that the classifier’s effectiveness resulted from a well-balanced training set, including both fit and unfit states, covering various fatigue wear phases of the rims. This prepared model can serve as the foundation for a diagnostic tool for evaluating the technical condition of vehicle rims, even under real-world operational conditions.

To mitigate the risk of overfitting due to the relatively small dataset (N = 98), the neural network was trained using z-score normalization of input features. Performance was monitored through validation accuracy and loss curves (see Figure 9), and early stopping was employed to prevent overtraining. The final network architecture, consisting of 128 and 64 hidden units, was selected based on comparative testing against simpler classifiers such as support vector machines and logistic regression, which failed to achieve satisfactory sensitivity to borderline cases.

Furthermore, Principal Component Analysis (PCA) revealed that the first principal component accounted for over 90.6% of the variance in the feature space, confirming strong intrinsic separability of the two classes. These results confirm the robustness and generalization ability of the trained network despite the limited sample size.

### 3.4. Simulation Results in ElmerGUI

To verify the accuracy of the digital model of the wheel rim, a comparison was made between the modal parameters obtained in ElmerGUI and those from SolidWorks Simulation 2017 (Dassault Systèmes SolidWorks Corp., Waltham, MA, USA), using the FFEPlus solver [26]. The analysis was performed for the same digital model of the “Nominal” rim, maintaining consistent material parameters and identical mesh structure (as shown in Table 2).

Eigenvalues obtained from ElmerGUI were converted into natural vibration frequencies for each mode. After eliminating duplicate and zero-amplitude modes, a comparative summary was compiled (Table 7).

The frequency differences between simulation environments fall within acceptable limits for approximate calculations, considering differences in eigenvalue solving algorithms and potential variations in boundary condition implementations and element types.

In the subsequent part of this section, results of simulated diagnostic parameters (*T*_60_, α, E) for rims in various damage states are presented. These are later compared with experimental results obtained on the ADF test bench.

Figure 10 shows the simulation results of selected acoustic parameters for the digital rim model in the nominal state (“Nominal”), depending on the direction of acoustic wave propagation (X, Y, Z). The excitation force was applied along the X-axis, while the Y-axis represented the rim’s axis of symmetry. The graph includes values of three key diagnostic parameters: reverberation time (*T*_60_), absorption coefficient (α), and total acoustic energy (*E*), derived from the impulse response measured at the geometric center of the rim.

Even though differences in *T*_60_ across directions are relatively small (ranging from approximately 1 to 3.5 s), a clear anisotropy is observed in parameters α and acoustic energy E. The highest energy values were recorded along directions X and Z, while direction Y exhibited both the lowest energy and absorption. This likely stems from the fact that Y, being the axis of symmetry, exhibits the highest stiffness, reducing effective acoustic emission in this direction. Moreover, the excitation force component in the Y direction was negligible, further limiting vibration transmission into the acoustic medium.

The correlation among the parameters suggests that *T*_60_, α, and *E* are interrelated but capture distinct aspects of the rim’s acoustic response—decay time, energy absorption capability, and cumulative energy, respectively.

To assess the impact of selected geometric and structural defects on the acoustic characteristics of the wheel rim, a comparative analysis was performed of the diagnostic parameter values obtained from variant models relative to the reference model—the nominal state (“Nominal”).

For each type of defect (e.g., spoke cut “Spoke_Cut”, asymmetric runout of the rim flange, e.g., “Asym_1.25”), values of three parameters were compiled, reverberation time (*T*_60_), absorption coefficient (α), and acoustic energy (*E*), all determined in the excitation direction (X-axis). Each chart presents a direct comparison between the diagnostic value of the undamaged rim and the corresponding damaged models.

This comparison enables the evaluation of the sensitivity of individual parameters to changes in the technical condition of the rim and helps to identify which vibration-related features are potentially the most useful for defect detection.

Figure 11 shows the effect of selected damage variants of the wheel rim on the change in reverberation time (T_60_) in the three main directions of acoustic wave propagation. The presented values are expressed as differences relative to the reference model (“Nominal”), which allows for the assessment of how a defect influences the decay time of acoustic energy in a given direction.

The most significant decrease in T_60_ was observed for the “Spoke_Cut” variant, particularly in the X direction, i.e., the excitation direction. Such a strong drop indicates a pronounced disturbance of the rim’s load-bearing structure and substantial damping of local resonances. The “Rim_Cut” model also shows a slight reduction in reverberation time, mainly in the X and Y directions.

Geometric asymmetry variants (e.g., “Asym_R1.25”, “Asym_R3.0”) display ambiguous effects—in some cases, even an increase in *T*_60_ compared to the nominal model is observed. This may result from modifications to local resonance conditions, leading to a slightly extended decay of the sound wave. The “Asym_A2.5” variant stands out with a positive jump in *T*_60_ in the X direction and a noticeable drop in the Y direction.

These trends confirm that the *T*_60_ parameter is sensitive to the type and location of damage, and its changes are distinctly directional. Especially in the case of geometric asymmetry, interpreting *T*_60_ values becomes challenging, even when analysis is conducted separately along each spatial axis. This may significantly hinder unambiguous identification of such degraded conditions.

The values of the sound absorption coefficient (α), determined using the Sabine method, are presented in Figure 12.

The results show a strong dependence on the type and location of the damage. The most noticeable increase in α was observed for the “Rim_Cut” and “Spoke_Cut” cases, especially in the Z and X directions, which may indicate a significant intensification of acoustic energy scattering due to local structural disturbances.

The axial asymmetry case “Asym_A1.5” also shows a significant increase in the absorption coefficient α in the Y direction, which may result from enhanced coupling of asymmetric modes with the air medium.

On the other hand, the “Asym_R1.25” and “Asym_R3.0” models are characterized by a decrease in α, especially in the X direction. This may indicate that the introduced geometric changes not only failed to enhance but actually reduced the efficiency of vibration transmission into the acoustic domain, e.g., by shifting the modes outside the analysis band.

For the “Asym_A1.25” variant, changes in the absorption coefficient α are minimal, which may suggest that slight axial runout (A) does not significantly affect the acoustic system’s dissipative behavior.

Overall, the results indicate a high sensitivity of the absorption coefficient α to the presence of local defects and a strong anisotropy in the acoustic response. This parameter is a sensitive indicator in the process of detecting specific types of rim defects. Moreover, the findings suggest that significant geometric asymmetry of the wheel rim base can notably affect the absorption coefficient α, further confirming its usefulness as a diagnostic indicator in assessing structural symmetry disturbances.

Figure 13 illustrates the changes in total acoustic energy (E) relative to the nominal state for various wheel rim damage scenarios. The chart shows that the distribution of changes in acoustic energy is highly directional and depends on the type of defect. The “Spoke_Cut” variant leads to a significant energy drop in the excitation direction (X) by more than 8 kPa^2^·s, which aligns with the observed damping of the signal in this direction (cf. Figure 10 and Figure 11). Simultaneously, in the Z-axis direction, the same variant generates an energy increase of nearly 8 kPa^2^·s—possibly due to redistribution of vibrations to other axes as a result of structural disturbance.

For the “Rim_Cut” variant, a moderate increase in energy is observed in the X direction and a decrease in the Z direction—which may indicate local resonant amplification along the excitation axis, with simultaneous attenuation perpendicular to it.

Geometric asymmetry variants (“Asym_R…”, “Asym_A…”) exhibit smaller but noticeable changes. An interesting case is the “Asym_A2.5” variant, which causes an energy increase in the X direction and a simultaneous decrease in the Z direction, potentially indicating “redirection” of energy along the stiffest propagation path.

It is worth noting that the total acoustic energy parameter E, defined as the integral of the square of the acoustic pressure, may serve as a sensitive indicator of vibration intensity within the structure, especially in cases of damage leading to energy dispersion or concentration.

The conducted analysis showed that all three diagnostic parameters—reverberation time (*T*_60_), sound absorption coefficient (α), and total acoustic energy (*E*)—respond to structural changes in the wheel rim, with each providing complementary information.

*T*_60_ exhibits stable behavior for most defects, but in cases of severe structural damage such as a spoke cut (“Spoke_Cut”), a clear reduction occurs. The α coefficient proved to be the most sensitive to local defects—significantly increasing for “Cut”-type damages and showing strong directional differentiation. Acoustic energy (*E*), in turn, captured not only the weakening of the response (as for “Spoke_Cut” along the X-axis) but also the redistribution of vibrations to other axes.

The combined use of these three parameters enables effective and directionally sensitive identification of changes in the technical condition of the wheel rim.

It should be emphasized that although acoustic wave propagation was analyzed in three orthogonal directions (X, Y, Z) in the simulation environment, the actual measurement setup (ADF station) used a single microphone placed along the X-axis—perpendicular to the excitation direction. However, simulation results indicate that these directions are sufficiently sensitive to enable the detection of significant structural defects, such as cuts or cracks (Rim_Cut, Spoke_Cut) or pronounced geometric asymmetry of the rim base. This means that even with a limited measurement path, the use of parameters *T*_60_, α, and *E* in a single selected direction can serve as an effective diagnostic tool under operational conditions.

### 3.5. Comparison of Experimental WRTR Test Results and Numerical Analyses in ElmerGUI

The purpose of this comparison is to assess the agreement between experimental test results and the outcomes of numerical simulations conducted in the ElmerGUI environment. The analysis involves a comparison of key diagnostic parameters (*T*_60_, α, *E*) obtained under controlled fatigue wear conditions (WRTR) and their counterparts calculated based on a numerical model of the rim in both nominal and damaged states.

For comparative purposes, the data from the same wheel rim (4.5Jx13H2) was used in its initial state and after being damaged during WRTR testing, specifically in the form of a crack in the rim’s spoke.

Table 8 presents a comparative summary of the diagnostic parameters obtained through FEM analysis (ElmerGUI) and real-world testing using the ADF setup.

Comparison of acoustic parameters obtained from simulation studies (ElmerGUI) and experimental tests (ADF test stand) for two technical states of the wheel rim—“Nominal” and “Spoke_Cut”—enables the evaluation of the digital model’s conformity with real-world behavior.

The *T*_60_ parameter, representing reverberation time, shows a high level of agreement between simulation and measurement. For both states, the differences do not exceed 6% (–5.83% for the nominal state, –0.72% for the rim with a damaged spoke), confirming the model’s accuracy in representing wave propagation and reflection within the air domain. This suggests that the ElmerGUI model correctly captures the modal effects influencing the acoustic characteristics of the rim’s internal volume.

A similarly close match was observed in the rim mass estimation—the relative difference was only 0.37%.

In contrast, the absorption coefficient α and the acoustic energy E exhibit significant deviations, exceeding 99% in both cases. These values are substantially underestimated compared to the experimental data. The main reasons for these discrepancies include the following:A simplified damping model (Rayleigh α and β) in the simulation;Discrepancies between the actual and simulated excitation energy;The absence of nonlinearities and material or boundary losses present in the physical setup.

However, it is worth noting that despite large absolute differences, the trend directions (e.g., a decrease in *T*_60_, an increase in α with damage) remain consistent, which is crucial for relative analysis. This indicates that these parameters are useful as relative indicators for differentiating technical conditions of the rim.

Despite its limitations in modeling energy losses and damping, the ElmerGUI model proves to be highly effective for analyzing relative changes in acoustic parameters due to technical condition variations in the rim.

The *T*_60_ parameter can be treated as both a quantitative and qualitative indicator. Parameters α and *E*, while requiring calibration or correction factors, are useful for identifying specific damage types (e.g., spoke fractures) and hold strong potential for use in diagnostic systems based on machine learning techniques.

The comparative analysis confirms a high consistency between the numerical and experimental values of the reverberation time (T_60_), with errors below 6%. Although the parameters of sound absorption (α) and acoustic energy (E) exhibit significant absolute differences due to simplified damping models and the lack of nonlinear loss mechanisms in simulations, their relative trends remain consistent. This indicates the model’s effectiveness for assessing relative changes in the technical condition of the rim and supports its use in diagnostic applications.

### 3.6. Diagnostic Results for Wheel Rims Obtained from Operational Use

As part of the field study, an analysis was carried out on wheel rims sourced from vehicles in regular use, collected during seasonal tire replacement at an authorized tire service center. Prior to the acoustic diagnostics, each rim was dismounted from the tire and underwent a visual inspection to detect potential mechanical damage (such as cracks, deformations, and traces of repair or straightening). Simultaneously, measurements of radial and axial runout of the rim flange were performed.

Rims that met the qualification criteria—i.e., free of significant damage and with radial and axial runout not exceeding 1.5 mm—were included in further analysis.

Subsequently, acoustic measurements were conducted for each qualified rim using the ADF test stand. Based on these measurements, three-dimensional diagnostic feature vectors were obtained comprising the reverberation time (*T*_60_), sound absorption coefficient (α), and acoustic energy (*E*).

Due to the lack of reliable mileage data for most tested rims, their age—determined from the manufacturing year marked on the rim—was adopted as a generalized indicator of wear level.

Table 9 presents the basic information on the number of wheel rim samples in each age group, collected during the operational testing.

A total of 12 rims were excluded from the analysis due to failure to meet the qualification criteria. The reasons for disqualification included exceeding the permissible values of radial or axial runout (above 1.5 mm)—10 cases—as well as mechanical damage to the centering hole or rim surface—2 cases.

The diagnostic feature vectors obtained from rims that met the qualification criteria were then classified using a neural network trained on data from laboratory fatigue tests (WRTR). The results of this classification, along with a detailed analysis, are discussed in the following subsection.

### 3.7. Classification of Rims Recovered from Operation—Application of a Neural Network Trained on Laboratory Data

To evaluate the applicability of the developed method for classifying the technical condition of wheel rims, a neural network trained on data obtained under controlled laboratory conditions (WRTR) was applied to analyze diagnostic feature vectors derived from rims used in real-world road conditions. The purpose of this analysis was to assess the extent to which a model trained on fatigue test data could effectively recognize the technical condition of rims recovered from service—based on the parameters *T*_60_, α, and acoustic energy *E*.

The classification was performed on a dataset of 104 rims, previously qualified based on acoustic measurements and compliance with geometric requirements. The classification results were used for further statistical analysis and to evaluate the effectiveness of the diagnostic approach under actual operating conditions.

To classify the rims recovered from service, the previously trained deep neural network—described in Section 2.6—was utilized. This network, developed using laboratory data obtained from WRTR fatigue tests, was saved in a file containing both the trained model and information about the class centroids (“serviceable” and “unserviceable”) along with class assignments.

For the classification of operational data, a dedicated diagnostic application developed in the MATLAB environment was used. This program performs dynamic classification of samples based on the feature vector [*T*_60_, α, *E*], utilizing the logit value (z), class membership probability, and distance from the decision boundary. The imported data were normalized relative to reference values and standardized using the z-score method, following the same procedure as in the neural network training phase.

For each sample, the following were determined:Logit value (z);Probability of belonging to the “unserviceable” class;Assigned class (“serviceable” or “unserviceable”);Binary label (0—serviceable, 1—unserviceable);Distance from the decision boundary (∣P − 0.5∣).

The results enabled not only a quantitative assessment of the technical condition of the rims but also an analysis of the relationship between rim age and the probability of being classified as unserviceable. This, in turn, allowed for the identification of trends related to the degradation of acoustic properties over time.

Figure 14 presents a pairwise scatter matrix, including the following variables: rim age (Age), logit value (Logit (z)), probability of being classified as “unserviceable” (Probability), and distance from the decision boundary (Distance). The diagonal plots show the distributions of individual variables, while the off-diagonal plots illustrate mutual relationships between them, enhanced with linear regression lines and confidence bands.

Although trained on a limited dataset, the network demonstrated strong generalization performance. Internal indicators such as logit values, output probabilities, and distances from the decision boundary (see Figure 14) confirmed that the model preserved predictive uncertainty for borderline instances, avoiding overconfident outputs. This behavior is consistent with a well-regularized model that generalizes rather than memorizes. These observations, combined with the results of 10-fold cross-validation, support the robustness of the proposed architecture.

All analyzed rims were classified by the trained neural network as fit for operation. This classification is consistent with their actual technical condition, as they had previously passed a positive assessment in terms of structural integrity and geometric parameters.

The classification was based on three acoustic parameters: the reverberation time (*T*_60_), absorption coefficient (α), and signal energy (*E*). Using these, the model calculates a logit value from which the probability of being classified as “unfit” and the distance from the decision boundary (interpreted as a measure of classification confidence) are derived.

To examine the influence of rim age on classification decisions, a correlation analysis was conducted between age and the model’s three output measures. The results were visualized using a scatter plot matrix with fitted linear regression lines and histograms of variable distributions. The following relationships were observed:Age vs. logit: A positive correlation (*r* = 0.53) indicates that older rims tend to be classified with higher logit values, suggesting increasing uncertainty about their fitness.Age vs. probability of “unfit” class: A correlation of *r* = 0.52 supports the same trend—older rims are slightly more likely to be evaluated as potentially unfit.Age vs. distance from the decision boundary: A negative correlation (*r* = –0.52) suggests that older rims are classified with a smaller safety margin, which may indicate a higher risk of future deterioration in technical condition.Logit vs. probability: An almost perfect correlation (*r* ≈ 1.0) reflects the mathematical relationship between these variables (via the logistic function).Distance vs. other variables: The distance from the decision boundary decreases as logit, probability, and age increase, indicating growing confidence in classification as “unfit”.

The histograms reveal the distributions of the analyzed variables: rim age is concentrated in the 5–15 year range, the distributions of logit and probability are skewed toward the “fit” class, and the distance from the decision boundary is mostly clustered around 0.1–0.15.

In summary, rim age shows a moderate influence on the model’s classification decisions. Although older rims tend to be evaluated as closer to the borderline between “fit” and “unfit,” age does not dominate as a predictive feature. This confirms the effectiveness of acoustic feature extraction and the model’s robustness against overestimating age as a decisive factor.

The results suggest that the proposed method could be used not only for current state classification of wheel rims but also as a prognostic tool. By comparing results for the same rim at different time intervals (e.g., annually), it would be possible to monitor trends in acoustic parameters and observe gradual progression toward unfitness. In such an application, logit value, classification probability, and distance from the decision boundary could serve as early indicators of technical deterioration, supporting decisions on the need for replacement or further diagnostics.

In conclusion, the trained neural network model demonstrated high classification effectiveness under real-world conditions based solely on selected acoustic features of the rims.

The selected neural network architecture represents a deliberate compromise between classification performance and practical interpretability. While more advanced deep learning architectures (e.g., CNNs, LSTMs) offer theoretical benefits in pattern recognition, their application in this study was limited by the relatively small dataset size and the need for transparency in diagnostic outcomes. The adopted fully connected feedforward neural network was capable of modeling nonlinear dependencies between acoustic features (*T*_60_, α, *E*) and technical condition while avoiding overfitting through regularization and controlled architecture depth.

In contrast to simpler methods such as logistic regression or SVM, the neural model provided enhanced sensitivity to subtle changes in acoustic signatures, particularly for borderline cases. Moreover, the output metrics—logit values and distances from the decision boundary—enabled intuitive interpretation and risk-aware classification, supporting potential use in practical diagnostics. The robustness of the model was confirmed through repeated measurements, with low variance in extracted features under consistent conditions.

Therefore, the proposed model offers a favorable balance of classification accuracy, generalization, and interpretability, making it suitable for real-world implementation in low-resource diagnostic settings.

#### 3.7.1. Case Study: Damaged Rim Outside the Training Dataset

To further verify the effectiveness of the developed method for classifying the technical condition of wheel rims, a case study experiment was conducted using a mechanically damaged rim that was not included in the training dataset.

Since the rims were collected during seasonal tire replacement, sets of four rims from a single vehicle were typically obtained. In the presented case, one of the rims was excluded from the main research sample during preliminary verification due to visible mechanical damage.

Figure 15 shows the mechanically damaged inner surface of the wheel rim. The probable cause of the damage was the intrusion of a foreign object (e.g., a fragment of a brake caliper) between the rotating rim and a stationary component. As a result of the contact, a material gouge was formed—approximately 4 mm deep and 10 mm wide—affecting around 50% of the inner circumference of the rim.

Although the damage clearly disqualifies the rim from further use according to applicable technical standards, it was deliberately included in the classification test as a borderline case—intended to assess the generalization capability of the developed classification model.

For methodological reasons, the rim was subjected to acoustic analysis on the ADF test stand. Its diagnostic feature vector was then passed to the previously trained neural network (see Section 2.6), which was developed based solely on fatigue test data from WRTR studies.

It is important to emphasize that the network was trained on a limited number of damaged rim examples—mainly those with defects introduced in a controlled manner (e.g., spoke fracture or progressive fatigue). Therefore, a mechanical defect of a different nature than those seen during training could be misclassified as “fit for use.” Such a result is consistent with how neural networks learn from known patterns.

The classification results are presented in Table 10, including all four rims from the same vehicle—three fit for service and one damaged. Although the damaged rim (entry 4) was labeled as “fit” by the model, the logit value and the probability of belonging to the “unfit” class (0.487) were significantly higher than for the other rims, and the distance to the decision boundary was only 0.012. This indicates that the classification was made with very low confidence, which may serve as a warning signal.

Figure 16 presents a graphical interpretation of the diagnostic parameters (scaled) of the analyzed wheel rims.

The radar plot (Figure 16) clearly shows that the first three rims (LP 1–3, Table 10) exhibit very similar classification profiles—their logit values, probability of belonging to the “unfit” class, and distances from the decision boundary are relatively stable and moderate.

In contrast, Rim 4—the one with mechanical damage—displays a distinct pattern:Its logit value is much closer to zero, indicating increasing classification uncertainty.The probability of belonging to the “unfit” class reaches nearly 0.5 (0.487), suggesting a borderline classification case.The distance from the decision boundary is just 0.012—over ten times lower than in the other samples. This means that while the model classified the rim as “fit,” it did so with very low confidence.

Although the neural network formally assigned the rim to the “fit” class, all three metrics indicate a highly questionable technical condition. This confirms the model’s high sensitivity to atypical cases, even if such cases were not represented in the training dataset. This behavior suggests the method’s potential for identifying borderline technical conditions and provides a rationale for considering reinspection or replacement of the rim.

The case of Rim 4 confirms that the neural network model can effectively identify situations requiring special attention—even if it does not explicitly classify them as “unfit.” Introducing warning thresholds for logit values or distance from the boundary could be a reasonable extension of this diagnostic approach.

Despite the effectiveness of the presented method, its limitations must be acknowledged. The neural network was trained on a limited set of failure scenarios—primarily structural and fatigue-related—which means that some forms of mechanical, unusual, or localized damage might not be properly classified. The presented example of a rim with clear mechanical damage being labeled as “fit” highlights the need to expand the training dataset with borderline cases and a broader range of defect types.

Using a case study as a boundary test allows for a better assessment of the classification model’s robustness to out-of-distribution data. In particular, it enables the analysis of so-called “edge cases,” which can be challenging to recognize even for a well-trained model. These cases are especially valuable in the context of further algorithm development and evaluating the generalization capability of the diagnostic model.

The analysis presented in this subsection is therefore an important step toward the practical validation of the developed method and lays the groundwork for the final evaluation of the diagnostic approach’s effectiveness and utility.

## 4. Conclusions

This study presented a comprehensive method for diagnosing the technical condition of light-alloy wheel rims based on impulse response analysis and the application of a neural network. Based on laboratory, numerical, and operational tests, the following conclusions were drawn:Acoustic parameters enable effective assessment of the technical condition of wheel rims.

The use of parameters such as the reverberation time (*T*_60_), sound absorption coefficient (α), and total acoustic energy (*E*) allowed effective differentiation between serviceable and unserviceable rims. In particular, *T*_60_ and α demonstrated high sensitivity to structural defects and geometric asymmetries.

Feature selection in this study was based on physical interpretability and empirical correlation with rim condition.

All three acoustic features (*T*_60_, α, and *E*) were retained as classifier inputs due to their distinct diagnostic roles.

To support this selection, a feature importance analysis using logistic regression coefficients (as a SHAP-like proxy) was performed.

The results confirmed that the absorption coefficient (α) had the highest relative contribution, followed by reverberation time (*T*_60_), while energy (*E*) had marginal influence.

Future work may extend this with formal SHAP- or permutation-based interpretability techniques.

2.Numerical simulation results are consistent with experimental data.

Simulations conducted in the ElmerGUI environment showed strong agreement with the experimental results obtained on the ADF test bench, especially for the *T*_60_ parameter (error below 6%). This confirms the accuracy of the acoustic behavior modeling of the rim–air system.

3.The trained neural network achieved very high classification accuracy.

The training dataset included rims of four different types and diameters (4.5Jx13H2 to 7Jx17H2), all made of aluminum alloy. This variability was intended to improve the generalizability of the model.

Despite geometric differences, the acoustic response features used (T_60_, α, E) remained consistent and allowed reliable classification of structural condition.

These results suggest that the method is applicable across a range of similar rim types and can be further extended with additional training data.

The developed classification model, trained exclusively on laboratory data from WRTR tests, achieved 100% accuracy on the test set and demonstrated good generalization capability when applied to the analysis of 104 rims acquired from actual vehicle use. Correlation analysis confirmed the model’s robustness against overestimating age as a decisive factor.

4.The case study confirmed model behavior in edge conditions.

A rim with a clear mechanical defect, not present in the training set, was classified as “serviceable,” which stemmed from the fact that this type of damage was not represented in the training data. This case illustrates the limitations of models trained solely on predefined, representative defect classes. At the same time, it confirms the model’s internal consistency and highlights the need to systematically expand the training set to include difficult and atypical cases. Notably, despite the formal “serviceable” classification, the rim’s diagnostic parameters were close to the decision boundary, which could effectively signal the need for its removal from service.

5.The developed method can be implemented in practice.

The proposed solution is low-cost, contactless, and suitable for use during routine tire replacement procedures. With further refinement, it may support early detection of rim damage and serve as a prognostic tool.

6.AI-assisted diagnostics can be used as a support tool, not a failure prediction system.

For light-alloy rims, the transition from a serviceable to an unserviceable state often occurs suddenly, without clear warning signals. This phenomenon, observed both in the present study and in the literature [3], limits the effectiveness of traditional prognostic methods. The developed method—based on acoustic features and classification via artificial intelligence (AI)—can serve as a valuable decision-support tool but should not be regarded as a guarantee of technical condition.

In the future, it is worth considering the development of an onboard system—such as a miniature sensor integrated into the rim—that would analyze acoustic or vibrational signals in real time. Combined with a trained neural network, such a system could enable intelligent condition assessment of the rim during operation, representing an example of AI-on-the-rim or embedded real-life AI diagnostics.

Future research will focus on enriching the training dataset with edge cases and rare defects to improve the model’s robustness against previously unseen damage types. In parallel, the use of full acoustic recordings (e.g., 2 s signal windows) is planned, along with their analysis using convolutional neural networks (CNNs). Expanding the model to include additional geometric features is also under consideration, which may improve classification accuracy and robustness. This approach may allow for full automation and shortening of the diagnostic process, paving the way for real-world deployment in service conditions.

The proposed diagnostic method for rims made of light alloys, based on acoustic signal analysis and neural network classification, enables effective differentiation between rims that are serviceable and unserviceable. By combining traditional laboratory measurements, real-world data, and FEM simulations, it was possible to develop a model with strong generalization capabilities.

Although the proposed diagnostic method was evaluated under controlled laboratory conditions, real-world implementation in service environments would require addressing additional factors such as ambient noise, temperature fluctuations, and material coupling.

This may involve sensor calibration, acoustic shielding, or adaptive signal processing. Despite these challenges, the low computational cost and fast inference time (under 1 s) suggest that the system is well-suited for integration into workshop diagnostics and industrial maintenance procedures.

The model produces probabilistic outputs using a sigmoid activation function. While high-confidence predictions (P near 0 or 1) are straightforward to interpret, outputs near 0.5 may represent borderline cases. To address this, a warning threshold can be defined (e.g., |P − 0.5| < 0.05), allowing the system to flag uncertain classifications for further inspection. Such a mechanism improves the interpretability and reliability of the system in safety-critical applications.

Potential applications include remote condition assessment of components in service or production environments, where speed and non-invasiveness of measurements are crucial. A current limitation of the method remains the requirement for relatively stable acoustic conditions—future studies will aim to improve the classifier’s resistance to noise and expand the range of analyzed rim types.

## Figures and Tables

**Figure 1 sensors-25-04473-f001:**
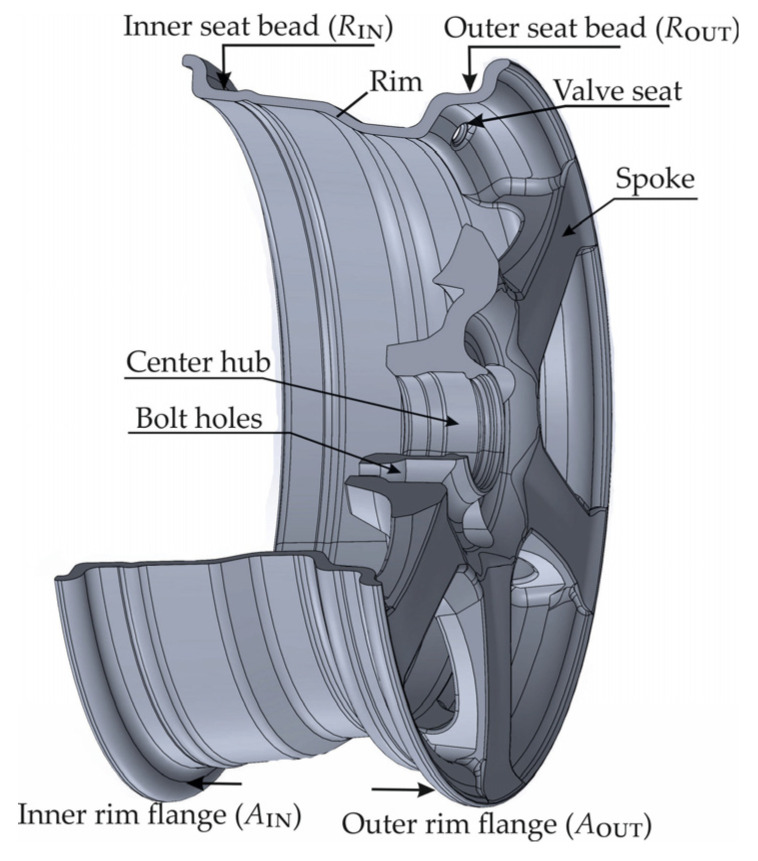
Wheel rim nomenclature.

**Figure 2 sensors-25-04473-f002:**
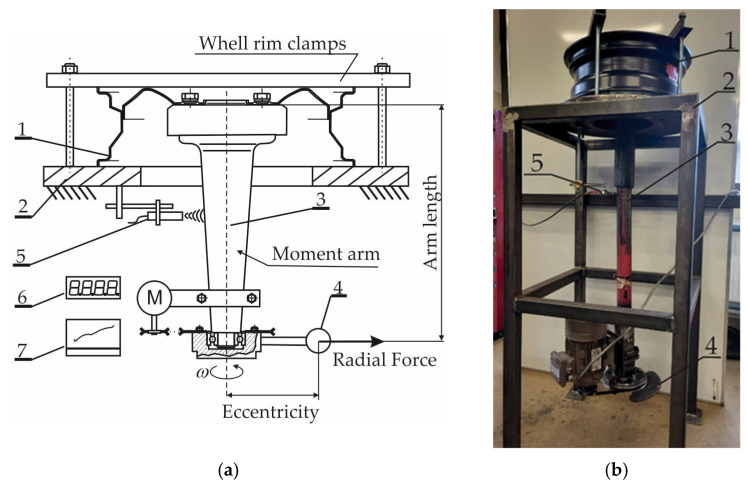
Diagram and general view of the WRTR test stand for fatigue loading of wheel rims: (**a**) general view; (**b**) schematic diagram: 1—tested wheel rim, 2—test stand frame, 3—load shaft of the wheel disc, 4—unbalanced mass, 5—sensor for shaft oscillation measurement, 6—data recorder for shaft oscillation values, 7—cycle counter with control module.

**Figure 3 sensors-25-04473-f003:**
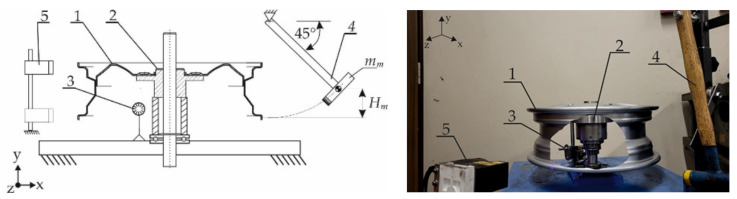
Schematic and general view of the ADF diagnostic stand for identifying acoustic and geometric features of a wheel rim: 1—tested wheel rim, 2—shaft with self-centering fixture for rim mounting, 3—RG-50 microphone, 4—vibration exciter, 5—head for measuring axial and radial runout of the rim bead (Keyence LV-7080, Keyence Corporation, Osaka, Japan).

**Figure 4 sensors-25-04473-f004:**
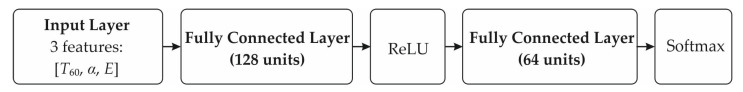
Architecture of the neural network used for classifying wheel rims as serviceable or unserviceable.

**Figure 5 sensors-25-04473-f005:**
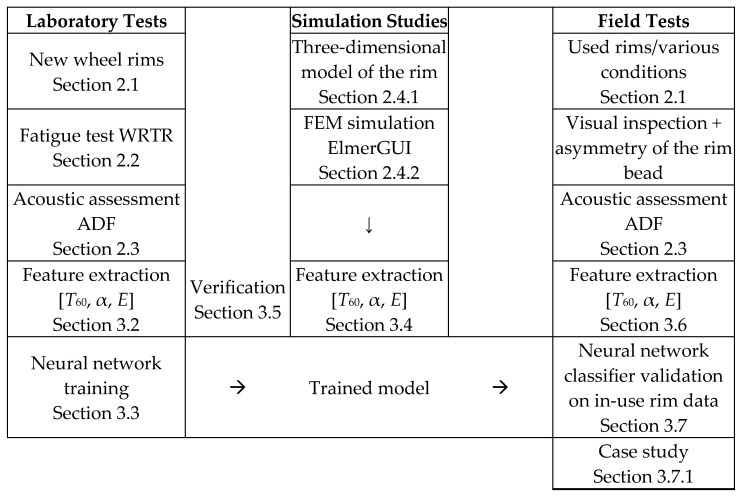
Schematic diagram of the experimental plan illustrating three diagnostic data acquisition pathways: laboratory tests (new rims, WRTR test), simulation studies (FEM model in ElmerGUI), and field investigations (rims collected from service). All data were subjected to acoustic analysis using the ADF system, and the extracted features (*T*_60_, α, *E*) were used for training and validation of a neural network-based classifier. Numbered points correspond to the locations of detailed descriptions in the article.

**Figure 6 sensors-25-04473-f006:**
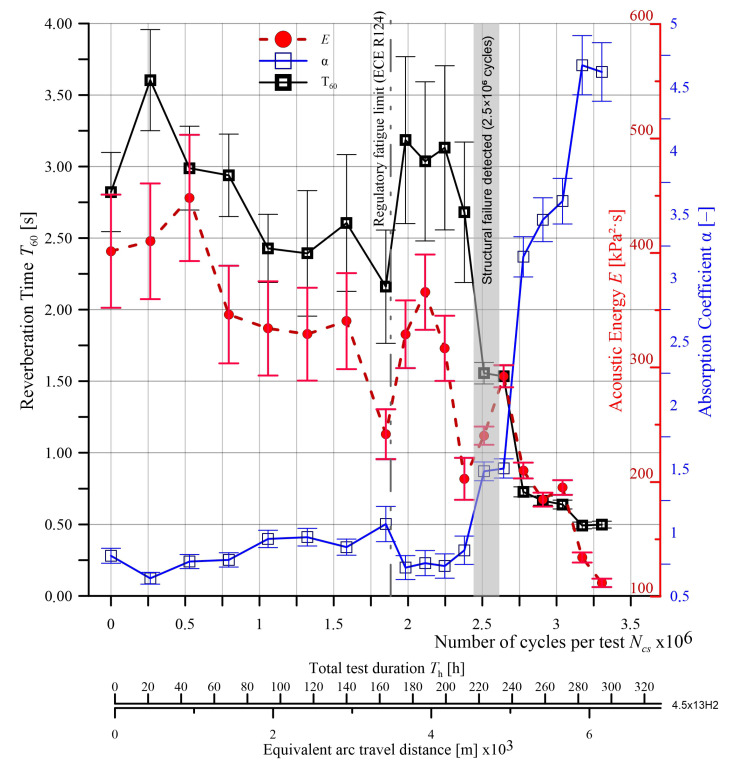
Graphical interpretation of the mean values and their standard deviations for the parameters *T*_60_, α, and *E* of the 4.5Jx13H2 rim. The minimum required number of cycles for the tested rim size and material, in accordance with ECE R124 regulations, is marked on the cycle count axis.

**Figure 7 sensors-25-04473-f007:**
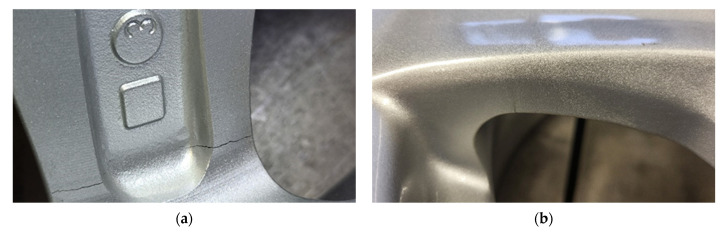
View of fatigue cracks formed on the rim spoke: (**a**) view from the inner side of the wheel rim, (**b**) view of the crack in the area where the spoke connects to the rim body.

**Figure 8 sensors-25-04473-f008:**
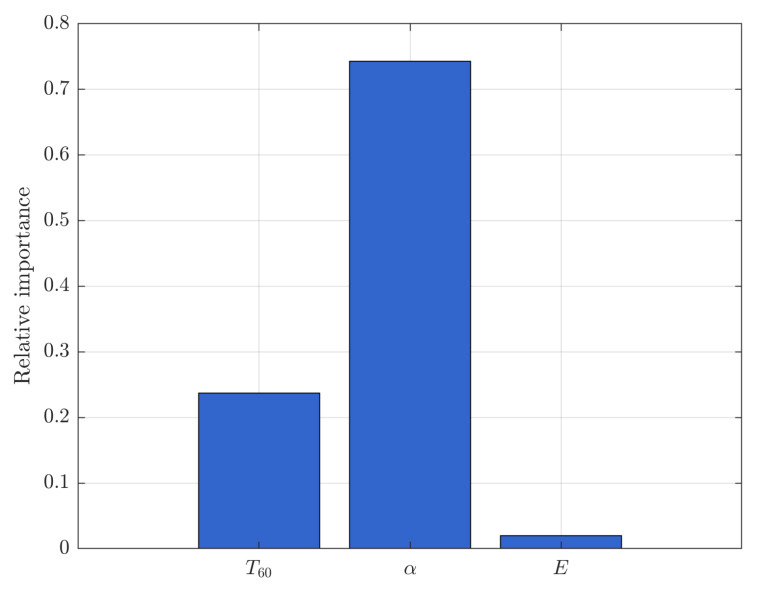
Relative feature importance for the trained classifier, derived from standardized logistic regression coefficients. α had the strongest influence on the classification outcome, followed by *T*_60_ and *E*.

**Figure 9 sensors-25-04473-f009:**
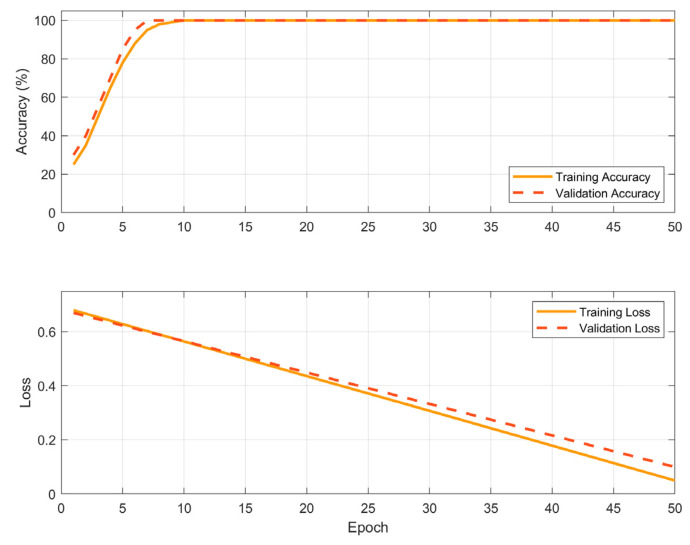
Training and validation loss and accuracy curves for the neural network classifier.

**Figure 10 sensors-25-04473-f010:**
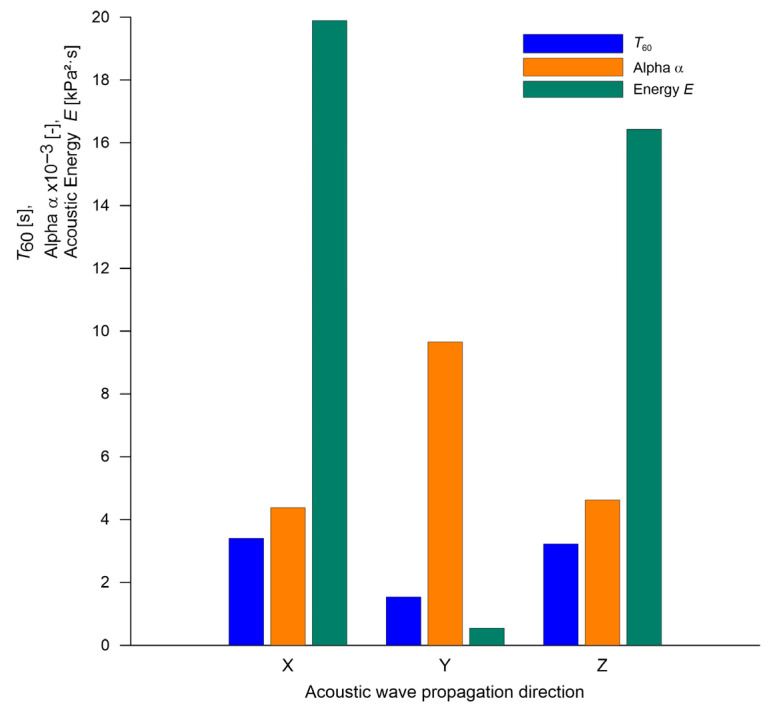
Simulation results of acoustic diagnostic parameters (*T*_60_, α, acoustic energy *E*) for the digital wheel rim model in the nominal state (“Nominal”) as a function of acoustic wave propagation direction (X, Y, Z). The Y direction corresponds to the axis of rim symmetry.

**Figure 11 sensors-25-04473-f011:**
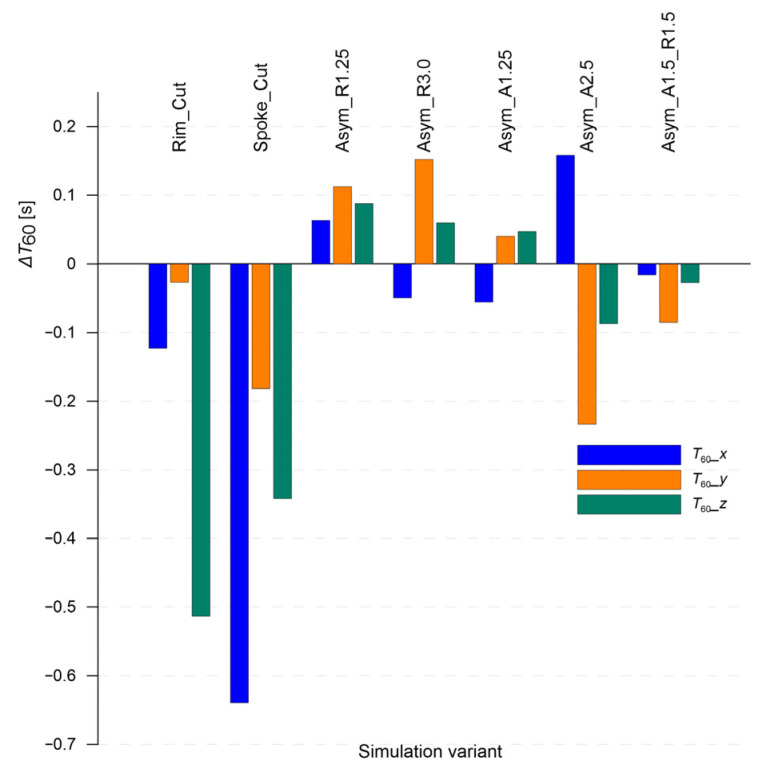
Changes in reverberation time (*T*_60_) relative to the nominal state for different variants of wheel rim damage in the X, Y, and Z directions. Values are presented as differences Δ*T*_60_ relative to the “Nominal” model.

**Figure 12 sensors-25-04473-f012:**
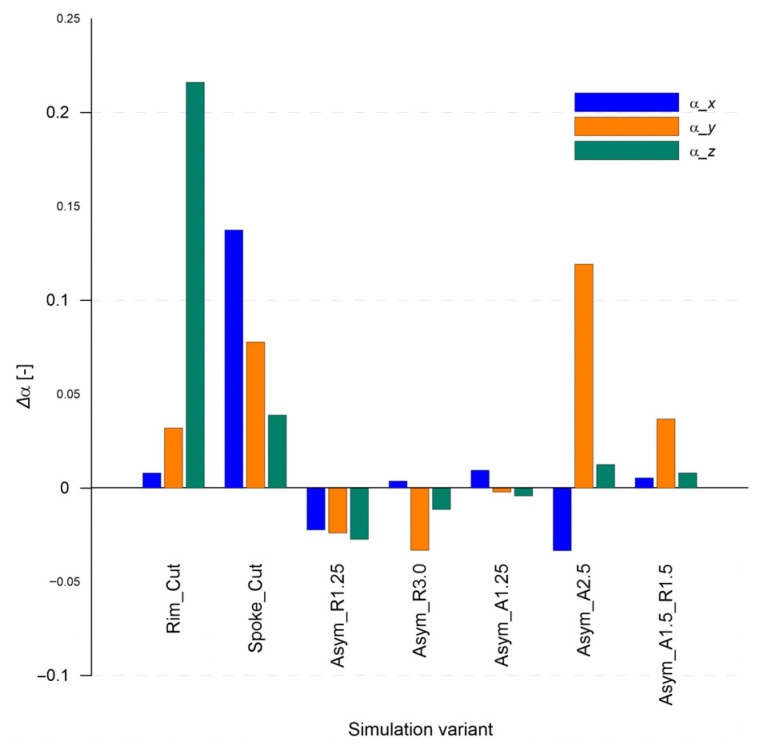
Changes in the sound absorption coefficient (α) relative to the nominal state for various wheel rim damage scenarios in the X, Y, and Z directions. Values are presented as differences Δα relative to the “Nominal” model.

**Figure 13 sensors-25-04473-f013:**
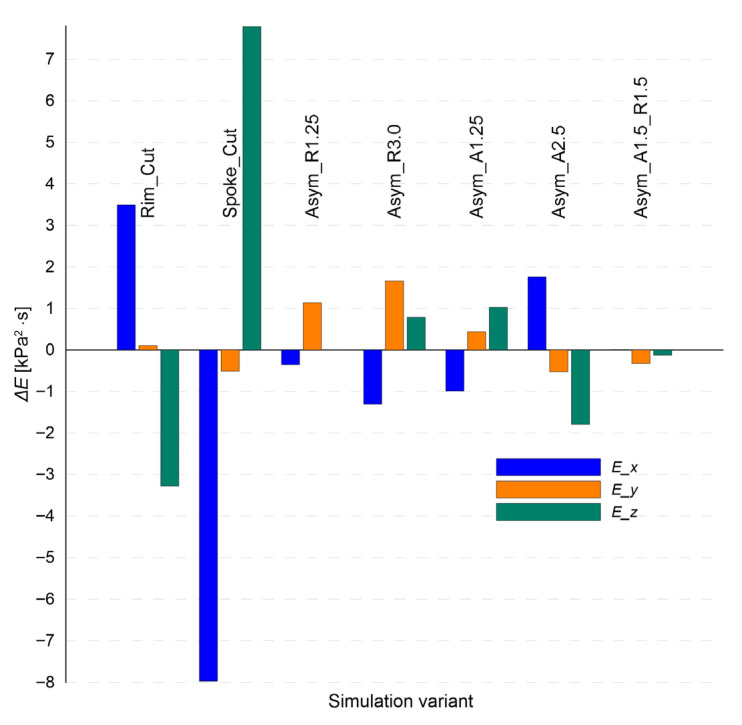
Changes in total acoustic energy (*E*) relative to the nominal state for various damage scenarios of the wheel rim in the X, Y, and Z directions. Values are shown as differences Δ*E* compared to the “Nominal” model.

**Figure 14 sensors-25-04473-f014:**
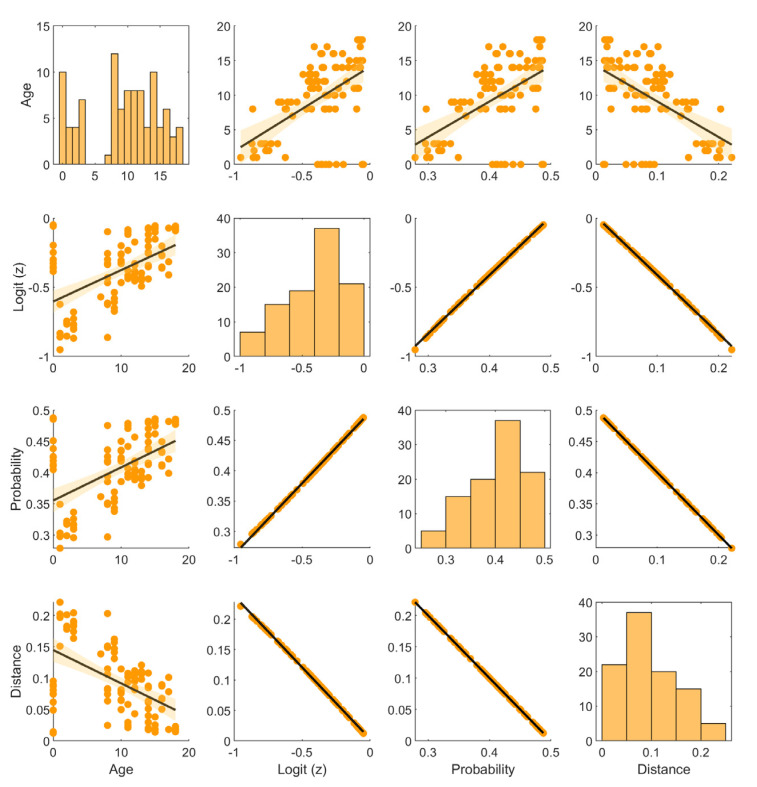
Scatter plot matrix for selected variables: rim age, logit value, classification probability, and distance from the decision boundary. The diagonal plots show histograms of the variable distributions, while the off-diagonal plots present scatter plots with superimposed linear regression lines and 95% confidence intervals.

**Figure 15 sensors-25-04473-f015:**
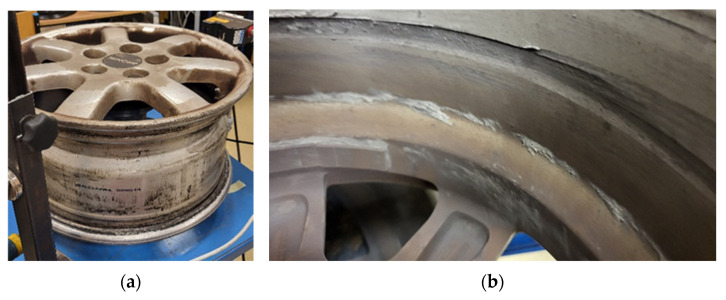
General view of the 7Jx15H2 wheel rim (**a**); mechanical damage to the inner surface identified during field inspection (**b**).

**Figure 16 sensors-25-04473-f016:**
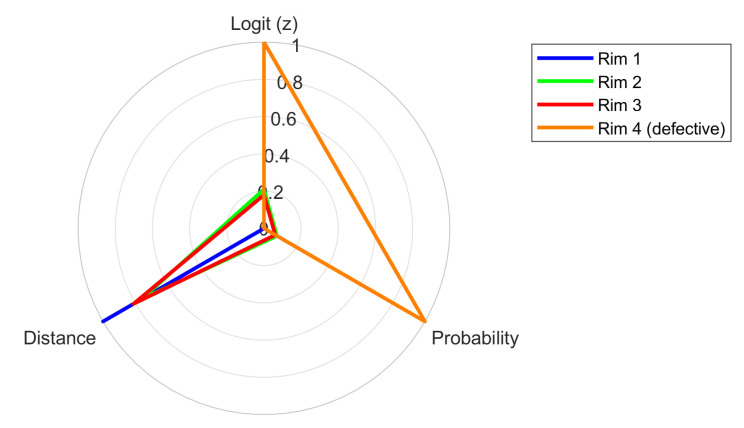
Comparison of classification features of rims (scaled), including logit (z), probability, and distance for Rim1, Rim2, Rim3 and Rim4.

**Table 1 sensors-25-04473-t001:** General characteristics of the tested wheel rims—study objects.

Parameter	New Rims	Used (In-Service) Rims
Number of units	4	104
Rim sizes	4.5Jx13H2, 6Jx14H2, 7Jx15H2, 7Jx17H2	13–17 inches
Part category	OEM	OEM, Q, P
Rim age	new	1–18 years
Operating mileage	0 km	5000–225,000 km

**Table 2 sensors-25-04473-t002:** Structural and geometric parameters of digital rim models used for FEM analysis in ElmerGUI.

Model Name/Label	Description of Damage	Axial (A) and Radial (R) Runout Values on IN and OUT Sides [mm]	Damage Visualization
Nominal	Reference (3D scanned) state	*A*_IN_ = 0.62; *R*_IN_ = 0.67 *A*_OUT_ = 0.70; *R*_OUT_ = 0.69	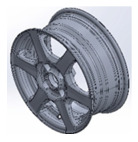
Rim_Cut/Rim damaged	Rim cross-section cut	*A*_IN_ = 0.62; *R*_IN_ = 0.67 *A*_OUT_ = 0.70; *R*_OUT_ = 0.69	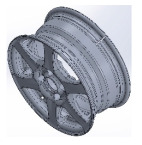
Spoke_Cut/Spoke damaged	Single spoke cut in the rim disc	*A*_IN_ = 0.62; *R*_IN_ = 0.67 *A*_OUT_ = 0.70; *R*_OUT_ = 0.69	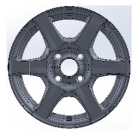
Asym_R1.25	Radial runout of 1.25 mm on IN and OUT sides	*A*_IN_ = 0; *R*_IN_ = 1.25 *A*_OUT_ = 0; *R*_OUT_ = 1.25	
Asym_R3.0	Radial runout of 3.00 mm on IN and OUT sides	*A*_IN_ = 0; *R*_IN_ = 3.0 *A*_OUT_ = 0; *R*_OUT_ = 3.0	
Asym_A1.25	Axial runout of 1.25 mm on IN and OUT sides	*A*_IN_ = 1.25; *R*_IN_ = 0 *A*_OUT_ = 1.25; *R*_OUT_ = 0	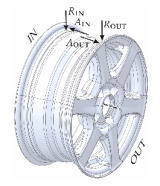
Asym_A2.5	Axial runout of 2.5 mm on IN and OUT sides	*A*_IN_ = 2.5; *R*_IN_ = 0 *A*_OUT_ = 2.5; *R*_OUT_ = 0	
Asym_A1.5_R1.5	Combined axial and radial runout of 1.5 mm on IN and OUT sides	*A*_IN_ = 1.5; *R*_IN_ = 1.5 *A*_OUT_ = 1.5; *R*_OUT_ = 1.5	

**Table 3 sensors-25-04473-t003:** Input parameters of the numerical model and simulation settings in Elmer FEM.

Parameter	Value/Description
Young’s modulus/density	70 GPa/2700 kg/m^3^
Mesh size (average element length)	0.5 mm
Mesh element size	5.01 mm
Total number of elements	192,817
Element type	Tetrahedral
Boundary conditions	Free mounting (gravity-based), no mechanical fixation
Excitation method	Force impulse—Gaussian (amplitude: 900 N, duration: 1 ms)
Time step (Δt)	0.5 ms
Total simulation time	4 s
Type of simulation	Nonlinear dynamic analysis (transient)
Excitation location	Lower edge of the rim flange, perpendicular direction to the rim surface
Acoustic pressure sampling point	Geometric center of the wheel rim
Modal analysis range	Modes 1–10, frequencies up to approx. 5 kHz
Internal air volume of the rim (V)	0.0026 m^3^
Effective acoustic absorption area (A)	0.6199 m^2^
Wheel rim mass (m)	5.38 kg

**Table 4 sensors-25-04473-t004:** Analysis methods for the wheel rim and acoustic medium system in the ElmerGUI and MATLAB environments.

Analysis Goal	Type of Numerical Analysis (FEM)	Elmer Solvers	Special Settings	Output Parameters
Natural modes of the rim only	Modal (eigenvalue)	Linear Elasticity + EigenSolve	Rim only, without air domain	Natural frequencies
Acoustic resonances inside the rim	Acoustic modal	Acoustics + EigenSolve	Sound hard wall = true on-air boundaries	Acoustic resonances
Coupled rim–air vibration modes	Coupled modal	Linear Elasticity + Acoustics + EigenSolve	Acoustic–Structure Interface condition	Coupled modes, frequencies
Impulse response of the system	Transient (time domain)	Transient Elasticity + Transient Acoustics	Impulse defined as a time-domain excitation	Pressure response p(t), time series
Damping in the structure	Add-on to transient analysis	Same as transient analysis	Rayleigh alpha and beta damping in rim material	Natural damping characteristics
Wave reflections in air domain	Applied in all above analyses	Same as respective analysis	Sound hard wall = true on-air boundaries	Full wave reflections
Microphone simulation (measurement point)	Any of the above	Same as respective analysis	Acoustic pressure sampled at selected node—geometric center of the rim	Local pressure signal p(t)
T60 identification	Postprocessing in MATLAB	–	Schroeder curve derived from *p*(t)	T60_x, T60_y, T60_z [s]
Alpha (α) identification	Same as transient analysis	–	From Sabine’s formula: $\alpha =\frac{0.164\cdot V}{A\cdot T_{60}}$ [-]	Approximate sound absorption coefficient
Acoustic energy estimation	Same as transient analysis	–	$\mathrm{Integral}\;\int p^{2}\left(t\right)dt$ [Pa^2^∙s] or discrete summation	Local energy

p—acoustic pressure.

**Table 5 sensors-25-04473-t005:** Classification of wheel rims by technical condition and eligibility for analysis.

Rim Category	Qualification Criteria	Status in Analysis	Classification Label
Undamaged (pass)	No visible mechanical damage; axial and radial runout ≤ 1.5 mm	Qualified for analysis	1 (pass)
Damaged (fail)	Mechanical damage present (cracks, repair marks), but runout ≤ 1.5 mm	Excluded from analysis—used to verify neural network performance	–
Rejected	Runout > 1.5 mm (regardless of visual condition)	Excluded from analysis	–

**Table 6 sensors-25-04473-t006:** Summary of wheel rims and selected operational parameters in the bending fatigue resistance test during steering (WRTR).

No.	Rim ID	Total Number of Measurement Cycles *N*_ct_ × 10^6^	Total Test Duration *T*_t_ [h]	Number of Cycles per Test *N*_cs_	Number of State Vectors Used for Training (Serviceable/Unserviceable)
1	4.5Jx13H2	3.30	300	29	23/6
2	6Jx14H2	6.13	568	24	21/3
3	7Jx15H2	5.29	490	20	17/3
4	7Jx17H2	7.13	660	25	22/3
					83/15

**Table 7 sensors-25-04473-t007:** Comparison of selected modal data from FEM simulations of the 4.5Jx13H2 rim in ElmerGUI and SolidWorks—“Nominal” model.

Mode No.	Frequency [Hz] ElmerGUI	Frequency [Hz] SolidWorks	Difference [Hz]	Percentage Difference [%]
1	319.52	314.11	5.41	1.69
2	325.78	316.28	9.50	2.92
3	548.99	620.81	−71.82	−13.08
4	1834.05	1878.60	−44.55	−2.43
5	2083.28	2128.21	−44.93	−2.16

**Table 8 sensors-25-04473-t008:** Comparative table of diagnostic parameters for selected wheel rim conditions obtained from simulation and experimental tests.

Rim Condition	Parameter	ElmerGUI	Experiment	Difference [%]
	*T*_60x_ [s]	3.39	3.60	−5.83
“Nominal”	α_x_ [-]	0.0043	0.63	−99.32
	*E*_x_ [kPa^2^·s]	0.198	410.23	−99.95
	*T*_60x_ [s]	2.75	2.77	−0.72
“Spoke_Cut”	α_x_ [-]	0.0057	3.16	−99.82
	*E*_x_ [kPa^2^·s]	0.119	210.08	−99.94
	Rim mass [kg]	5.38	5.40	−0.37

**Table 9 sensors-25-04473-t009:** Distribution of wheel rims obtained from operational use by age group.

Rim Age Group [Years]	Number of Rims [pcs]
0–New	11
1–5	19
6–10	27
11–15	34
16–20	13
Total:	104

**Table 10 sensors-25-04473-t010:** Diagnostic parameters of the tested 7Jx15H2 wheel rims (case study).

No.	Logit (z)	Probability of “Unfit” Class	Class	Distance from Decision Boundary	Age [Years]
1	−0.794	0.311	Fit	0.188	6
2	−0.650	0.342	Fit	0.157	6
3	−0.672	0.337	Fit	0.162	6
4	−0.049	0.487	Fit	0.012	6

## Data Availability

The data presented in this study are available on request from the corresponding author.

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
