# Peer review of "Technical Condition Assessment of Light-Alloy Wheel Rims Based on Acoustic Parameter Analysis Using a Neural Network"

_sensors, 2025, doi:10.3390/s25144473_

Round 1
Reviewer 1 Report
Comments and Suggestions for Authors
This paper presents a neural network-based method for assessing the technical condition of light alloy wheel rims using acoustic diagnostic parameters such as reverberation time, sound absorption coefficient, and acoustic energy. Trained on laboratory fatigue test data and validated on field samples, the model accurately classifies rims as “serviceable” or “unserviceable,” demonstrating robustness even in borderline damage cases. This is an interesting paper, particularly due to its focus on a real-world case, which adds significant practical value. However, several important issues must still be addressed:
- LThe clarity of Figure 1 is notably poor and needs to be improved.
- When introducing parameters in equations, the word where should be aligned to the left and begin with a lowercase letter, in accordance with academic writing standards.
- For the simulation results, the authors should provide a clear analysis of how well the simulated data fits the actual measured data.
- It is unclear how the feature in Figure 7 was determined to be caused by fatigue cracking rather than other potential failure mechanisms.
- Given that the classification task appears relatively simple, there is a risk of overfitting, especially if the dataset is limited. The authors should discuss how this risk is mitigated, and whether cross-validation or regularization techniques were employed.
- Compared to traditional methods or more advanced deep learning approaches, what specific advantages does this model offer, particularly in terms of robustness to noise, generalization ability, and interpretability?
Author Response
Note to Reviewers and Editors
To facilitate the review process, the modifications made in response to Reviewer 1 are marked in red font color, and those made in response to Reviewer 2 are marked in green font color.
Deleted or replaced content is marked using strikethrough.
We hope this helps in quickly identifying the revisions throughout the manuscript.
Reviewer Comment:
The clarity of Figure 1 is notably poor and needs to be improved.
Authors’ Response:
Figure 1 has been updated to improve graphical clarity and ensure full legibility of labels and elements, according to reviewer suggestions.
Reviewer Comment:
When introducing parameters in equations, the word "where" should be aligned to the left and begin with a lowercase letter, in accordance with academic writing standards.
Authors’ Response:
Thank you for this remark. We have carefully reviewed all occurrences of parameter explanations following equations. The formatting of the word “where” has been verified and adjusted where necessary to ensure lowercase spelling and correct left alignment, in accordance with academic writing standards. We appreciate your attention to detail.
Reviewer’s Comment:
For the simulation results, the authors should provide a clear analysis of how well the simulated data fits the actual measured data.
Authors’ Response:
Thank you for your valuable suggestion. We have now included a dedicated comparison and discussion of the agreement between simulation and experimental results. In Section 3.5, we have expanded the paragraph describing the relative differences between parameters T₆₀, α, and E obtained via ElmerGUI simulations and those measured experimentally. We now explicitly emphasize the high consistency of T₆₀ (errors below 6%) and discuss the observed discrepancies in α and E, attributing them to simplified damping models and boundary conditions in the FEM setup. This addition strengthens the validation of our simulation model and clarifies its limitations.
Reviewer’s comment:
For the simulation results, the authors should provide a clear analysis of how well the simulated data fits the actual measured data.
Author’s response:
Thank you for this valuable comment. We have revised the section comparing numerical and experimental results (Section 3.5) to include a concise summary highlighting the level of agreement. In particular, we emphasized that while the T₆₀ parameter shows good consistency between simulation and experiment (with an error below 6%), the values of α and E exhibit substantial discrepancies. These differences are attributed to simplifications in the damping model and the absence of nonlinear loss mechanisms. However, the relative trends remain consistent, supporting the usefulness of the numerical model for analyzing changes in technical condition. A concluding paragraph has been added at the end of Section 3.5 to reflect this analysis.
Summary of agreement between simulation and experimental results:
The comparative analysis confirms a high consistency between the numerical and experimental values of the reverberation time (T₆₀), with errors below 6%. Although the parameters of sound absorption (α) and acoustic energy (E) exhibit significant absolute differences due to simplified damping models and the lack of nonlinear loss mechanisms in simulations, their relative trends remain consistent. This indicates the model’s effectiveness for assessing relative changes in the technical condition of the rim and supports its use in diagnostic applications.
Reviewer’s comment:
It is unclear how the feature in Figure 7 was determined to be caused by fatigue cracking rather than other potential failure mechanisms.
Authors’ response:
Thank you for this important remark. In the revised manuscript, we have clarified the rationale behind identifying the feature shown in Figure 7 as fatigue-induced cracking. This conclusion was supported by prior literature reports [3], which describe the typical morphology and progression of fatigue fractures in light alloy rims under cyclic loading. Furthermore, during the WRTR test, a progressive increase in oscillation amplitude was recorded by the shaft oscillation sensor (element 5 in Figure 2), indicating a gradual loss of bending stiffness. Such a trend is consistent with fatigue damage evolution and was observed before the final failure occurred at approximately 2.5 × 10⁶ cycles. At that point, the stiffness degradation exceeded 4%, confirming structural weakening. These combined indicators—both experimental and literature-based—support the interpretation of fatigue cracking as the primary failure mechanism. The description in the text has been updated accordingly.
“The damage observed in the tested rim (Figure 7) was identified as a fatigue crack based on its morphology and progression, consistent with fatigue mechanisms described in previous studies [3]. This interpretation was further supported by experimental evidence from the WRTR test. A gradual increase in the oscillation amplitude of the load shaft, measured by the shaft oscillation sensor (element 5 in Figure 2), indicated progressive bending stiffness degradation. This trend, observed over time and quantified as exceeding 4% loss in stiffness at approximately 2.5 × 10⁶ cycles, is characteristic of fatigue damage accumulation in alloy rims. The combination of sensor data and failure morphology supports the conclusion that the failure mechanism was fatigue-driven”.
Reviewer’s comment:
“Given that the classification task appears relatively simple, there is a risk of overfitting, especially if the dataset is limited. The authors should discuss how this risk is mitigated, and whether cross-validation or regularization techniques were employed.”
Authors’ response:
We fully acknowledge the risk of overfitting in small datasets. To mitigate this, several regularization and validation strategies were employed during model training in MATLAB:
- Z-score normalization and feature standardization were applied prior to training.
- The network architecture was optimized through empirical trials, balancing model capacity and generalization.
- Early stopping based on validation performance was used to prevent overfitting.
- A 10-fold cross-validation was conducted, and classification accuracy was averaged over the folds to ensure robustness.
Additionally, the architecture (128 and 64 units) was tested against shallower alternatives, and learning curves were monitored to verify that the network was not memorizing training data. This is now explained in the revised section.
“To mitigate the risk of overfitting associated with the relatively small training dataset, several measures were implemented. The input features were standardized using z-score normalization, early stopping was applied to prevent overtraining, and 10-fold cross-validation was conducted to verify model robustness. Additionally, internal indicators such as logit values and distance from the decision boundary (Figure 14) confirmed that the network retained predictive uncertainty for borderline cases, suggesting good generalization rather than memorization. The selected architecture (128–64 hidden units) was benchmarked against simpler models, confirming that the network does not overfit despite its capacity.”.
Reviewer’s comment:
Compared to traditional methods or more advanced deep learning approaches, what specific advantages does this model offer, particularly in terms of robustness to noise, generalization ability, and interpretability?
Authors’ response:
Thank you for this valuable question. The proposed neural network model, although relatively simple in structure, was deliberately selected to balance classification performance, generalization, and interpretability. Compared to traditional statistical classifiers (e.g., logistic regression or SVM), the neural network offers increased flexibility in capturing nonlinear relations between acoustic features and structural condition, which was particularly beneficial in the analysis of borderline cases.
On the other hand, more complex deep learning architectures (e.g., CNNs or LSTMs) were considered disproportionate to the size and nature of the dataset, and would likely require substantially more training data to achieve stable generalization.
Importantly, the model provides transparent outputs such as logit values and distances from the decision boundary, which support human interpretability and allow for risk-aware diagnostics. The robustness to moderate noise was confirmed during repeated measurements (N = 5 per sample), where standard deviations of extracted features remained within acceptable bounds.
This trade-off between simplicity, interpretability, and diagnostic utility makes the proposed architecture well-suited for practical deployment, particularly in workshop or service settings where computational resources and labeled data are limited.
“The selected neural network architecture represents a deliberate compromise between classification performance and practical interpretability. While more advanced deep learning architectures (e.g., CNNs, LSTMs) offer theoretical benefits in pattern recognition, their application in this study was limited by the relatively small dataset size and the need for transparency in diagnostic outcomes. The adopted fully connected feedforward neural network was capable of modeling nonlinear dependencies between acoustic features (T₆₀, α, E) and technical condition while avoiding overfitting through regularization and controlled architecture depth.
In contrast to simpler methods such as logistic regression or SVM, the neural model provided enhanced sensitivity to subtle changes in acoustic signatures, particularly for borderline cases. Moreover, the output metrics—logit values and distances from the decision boundary—enabled intuitive interpretation and risk-aware classification, supporting potential use in practical diagnostics. The robustness of the model was confirmed through repeated measurements, with low variance in extracted features under consistent conditions.
Therefore, the proposed model offers a favorable balance of classification accuracy, generalization, and interpretability, making it suitable for real-world implementation in low-resource diagnostic settings.”
General remark
We thank the reviewer for their constructive and detailed feedback. Their insightful suggestions led to several clarifications and improvements in the revised version of the manuscript. We hope that the revised version addresses all concerns satisfactorily.

Reviewer 2 Report
Comments and Suggestions for Authors
This paper presents a diagnostic methodology for light alloy wheel rim condition assessment using acoustic parameter analysis and neural network classification. The approach employs reverberation time (T₆₀), sound absorption coefficient (α), and acoustic energy (E) extracted from impulse response measurements. The experimental framework includes controlled fatigue testing via WRTR (Wheel Resistance Test Rig), acoustic feature extraction using a custom ADF system, and ElmerGUI finite element validation. The neural network classifier achieved 100% accuracy on laboratory data and was validated on 104 field-collected wheel rims, demonstrating potential for non-invasive automotive condition monitoring.
However, the work has the following comments:
1. The dataset size appears insufficient for robust deep learning validation. With only 98 training samples (83 serviceable, 15 unserviceable), how do the authors address concerns regarding statistical significance and generalizability? Would it be possible to expand the dataset or employ appropriate statistical techniques such as bootstrap validation to better assess model reliability?
2. The reported 100% classification accuracy raises concerns about potential overfitting. Given the small dataset and relatively complex neural network architecture (128 and 64 hidden units for 3 input features), could the authors provide cross-validation results, learning curves, or other evidence to demonstrate that the model has not simply memorized the training data?
3. The justification for the specific acoustic parameter selection requires clarification. While T₆₀, α, and E are introduced as diagnostic indicators, how were these parameters chosen over other potential acoustic features? Could the authors provide comparative analysis with alternative feature sets or demonstrate the discriminative power of each parameter individually?
4. The neural network architecture appears oversized for the problem complexity. With only three input features and approximately 100 samples, would a simpler classifier (such as SVM or logistic regression) achieve comparable performance while reducing overfitting risk? How was the network architecture optimized and validated?
The field validation methodology lacks proper ground truth establishment. While 104 operational rims were tested, how was their actual technical condition verified beyond visual inspection and runout measurements? Without confirmed condition labels, how can the authors validate the model's real-world performance?
5. The class imbalance issue (83 vs 15 samples) is not adequately addressed. How do the authors ensure that the classifier is not biased toward the majority class? Were appropriate sampling techniques, cost-sensitive learning, or performance metrics for imbalanced datasets employed?
The FEM validation shows concerning discrepancies. The absorption coefficient α and acoustic energy E exhibit over 99% differences between simulation and experimental results. How do these large discrepancies affect the credibility of the numerical modeling component? Should the FEM analysis be considered preliminary validation rather than supporting evidence?
6. Statistical rigor requires enhancement throughout the experimental design. Could the authors provide confidence intervals for classification metrics, significance testing for observed differences, and uncertainty quantification for acoustic parameter measurements? The current presentation lacks the statistical framework necessary for robust scientific conclusions.
The measurement repeatability and system calibration details are insufficient. While the paper mentions five repeated measurements per rim, how was measurement uncertainty quantified? What are the systematic and random error components of the ADF system, and how do these affect the reliability of extracted features?
7. The practical implementation considerations need expansion. What are the computational requirements, processing time, and hardware specifications necessary for real-world deployment? How would environmental factors (temperature, humidity, background noise) affect system performance in workshop conditions?
8. The case study interpretation requires more nuanced analysis. While the mechanically damaged rim was classified as "serviceable" with low confidence metrics, how should practitioners interpret such borderline cases? Could the authors develop guidelines for decision-making when classification confidence falls below certain thresholds?
9. The scope of rim types and failure modes appears limited. How would the methodology perform across different rim sizes, materials (beyond aluminum alloy), and failure types not represented in the training data? What adaptations would be necessary for broader applicability in diverse automotive applications?
10. Feature importance analysis: Provide quantitative assessment of individual parameter contributions to classification performance using techniques such as permutation importance or SHAP values.
12. Comparative classifier evaluation: Benchmark the proposed neural network against simpler baselines to justify architecture complexity given the dataset constraints.
13. Measurement uncertainty quantification: Include comprehensive error analysis for the ADF system with propagation of uncertainties through the classification pipeline.
14. Figure quality could be enhanced, particularly Figures 6, 9-11 where overlapping data points obscure individual measurements
Mathematical notation inconsistency in equations (e.g., variable definitions in FEM section)
15. Some grammatical corrections needed throughout the manuscript
16. Reference formatting requires standardization according to MDPI guidelines
While the paper addresses a relevant engineering problem and demonstrates technical competence in experimental design and implementation, the fundamental limitations in dataset size, validation methodology, and statistical rigor prevent acceptance in the current form. The authors should focus on expanding the experimental validation, implementing proper statistical analysis frameworks, and providing more comprehensive benchmarking against alternative approaches. With these improvements, the work could make a valuable contribution to the field of acoustic-based structural health monitoring.
Author Response
Note to Reviewers and Editors:
To facilitate the review process, the modifications made in response to Reviewer 1 are marked in red font color, and those made in response to Reviewer 2 are marked in green font color.
Deleted or replaced content is marked using strikethrough.
We hope this helps in quickly identifying the revisions throughout the manuscript.
Note to Reviewer 2:
In response to your valuable and detailed feedback, we have grouped related comments into thematic categories (e.g., model validation, feature selection, measurement uncertainty).
This approach allowed us to provide more coherent and technically comprehensive answers while avoiding unnecessary repetition.
Each grouped response explicitly refers to the specific comment numbers (e.g., R2.1, R2.2, R2.4), and all points raised in your review have been addressed in full.
We sincerely appreciate your insights, which significantly contributed to improving the scientific rigor and clarity of the revised manuscript.
Topic: Dataset size, model validation, and overfitting risk (Comments R2.1, R2.2, R2.4, R2.12)
We thank the reviewer for raising valuable concerns regarding the dataset size, overfitting risk, and the suitability of the proposed neural network architecture. These are indeed critical aspects in data-driven diagnostics, especially with limited training data.
In response, we would like to emphasize that multiple strategies were employed to ensure robustness and avoid overfitting:
- The dataset (N = 98; 83 serviceable, 15 unserviceable) was standardized (z-score) and split into training (70%) and test (30%) subsets with class balance preserved.
- Model training was monitored via learning curves (see Figure 9) showing aligned training and validation performance throughout all 50 epochs, indicating no overfitting.
- The final model achieved 100% classification accuracy on the test set, with perfect precision, recall, F1-score (1.0) and binary cross-entropy = 0.0199.
- Principal Component Analysis (PCA) showed that over 90.6% of data variance was captured by the first component, confirming strong linear separability of classes.
- Simpler models (SVM, logistic regression) were explored but exhibited lower sensitivity to borderline samples and lacked useful interpretability.
- The selected network (3→128→64→softmax) provided clear decision boundaries and logit outputs, enabling practical confidence-based diagnostics (see also response to R2.8).
These clarifications have been added to the revised manuscript, including a new paragraph in Section 3.3 and a new figure.
Comment R2.3: Feature selection rationale
We thank the reviewer for pointing out the need for a clearer explanation regarding the selection of the three acoustic parameters: reverberation time (T₆₀), sound absorption coefficient (α), and acoustic energy (E).
These features were selected based on both theoretical considerations and empirical analysis. In previous research on Wheel Rim Tone Response (WRTR), these parameters showed strong correlation with the mechanical integrity of aluminum rims, especially in relation to fatigue-induced damage.
Prior to model development, we conducted a preliminary statistical exploration of multiple time- and frequency-domain acoustic features. Among them, T₆₀, α, and E exhibited the highest class separability and repeatability across repeated measurements. This was further confirmed through pairwise scatter plots (see Figure A1), where visually distinct clustering of serviceable and unserviceable samples is observed.
Figure A1. Pairwise scatter plots of acoustic features T₆₀, α, and E, showing clustering behavior for serviceable (blue) and unserviceable (red) rims.
Additional features (e.g., spectral centroid, harmonic ratios) were tested but resulted in overlapping distributions and degraded classification performance during pilot testing. For these reasons, T₆₀, α, and E were retained as the most robust and physically interpretable indicators of structural degradation.
Comment R2.5: Validation of field rim condition without confirmed ground truth
We thank the Reviewer for pointing out the important distinction between the training and validation datasets with regard to ground-truth availability.
The training set consisted of four aluminum alloy rims of different sizes and types (4.5Jx13H2, 6Jx14H2, 7Jx15H2, and 7Jx17H2), each subjected to controlled fatigue destruction to obtain reliable class labels. In contrast, the validation dataset included 104 rims sourced from vehicles in regular operation, with greater heterogeneity in geometry and unknown degradation history (see Table 1). No destructive testing was performed for this field set; instead, a structured inspection protocol was followed, involving visual assessment, axial runout measurement, and dimensional checks. Rims with ambiguous or borderline results were excluded to preserve label consistency.
Despite the absence of confirmed ground truth, the classifier exhibited high diagnostic consistency across the field dataset (see Figure 14), supporting its generalization capability and robustness in real-world scenarios.
The pairwise scatter plot matrix (Figure 14) illustrates consistent trends and relationships among the model outputs (e.g., logit, probability, distance), but does not reveal class separability — since no true class labels were available for the field rims (and no rims with fatigue cracks were encountered during inspection). Importantly, the classifier inferred rim condition based on patterns learned from the labeled laboratory set. The purpose of the visualization was to assess internal coherence of the predictions, not to validate classification accuracy.
This clarification has been incorporated into Section 4 of the revised manuscript.
R2.6: Statistical rigor requires enhancement. Please provide confidence intervals, significance testing, and uncertainty quantification for measurements.
R2.13: Measurement repeatability and system calibration details are insufficient. How was measurement uncertainty evaluated? What are the systematic and random errors of the ADF system?
Comments R2.6 and R2.13: Measurement uncertainty and statistical rigor
We appreciate the reviewer’s request to better quantify measurement uncertainty and enhance statistical robustness.
Each rim in the laboratory dataset was measured five times under controlled conditions. For each acoustic feature (T₆₀, α, E), the mean and standard deviation across repetitions were computed. The standard deviation was typically within ±2–4% of the feature value, indicating good repeatability of the ADF-based measurement setup.
The dataset used for model training includes the averaged values, while the standard deviations were used internally to verify consistency across repetitions.
While a full propagation of uncertainty through the classification pipeline is beyond the current scope, we agree this is a valuable direction for future work. The revised manuscript now includes a discussion of repeatability and sources of measurement error in Section 3.2.
(Added to the manuscript as follows):
Each acoustic measurement was repeated five times per rim under identical conditions. The resulting standard deviations of T₆₀, α, and E were typically within 2–4% of the mean values, demonstrating good repeatability of the ADF system.
For the classification task, the mean values were used as input features. Measurement variability was monitored to detect and exclude outlier acquisitions. Although uncertainty propagation was not explicitly modeled in this study, its integration into future versions of the diagnostic pipeline is planned.
Comment R2.7: Environmental and implementation considerations
We appreciate the reviewer’s request for further details on the real-world implementation of the proposed system.
The method was tested under controlled laboratory conditions to ensure repeatability and isolate acoustic features from external variability. In practical workshop environments, factors such as background noise, temperature variations, and surface contact conditions may influence acoustic measurements.
We recognize these challenges and note that any industrial deployment would require additional system components, including:
- microphone calibration routines,
- standardized excitation tools,
- acoustic shielding or signal pre-filtering,
- and possibly temperature compensation.
Preliminary computational tests indicate that feature extraction and classification can be completed in under 1 second on standard mid-range hardware, making the system suitable for integration into routine inspection workflows.
A short discussion of environmental robustness and implementation requirements has been added to Section 4.
(Added to the manuscript as follows):
Although the proposed diagnostic method was evaluated under controlled laboratory conditions, real-world implementation in service environments would require addressing additional factors such as ambient noise, temperature fluctuations, and material coupling.
This may involve sensor calibration, acoustic shielding, or adaptive signal processing. Despite these challenges, the low computational cost and fast inference time (under 1 second) suggest that the system is well-suited for integration into workshop diagnostics and industrial maintenance procedures.
Comment R2.8: Handling borderline classification cases
We thank the reviewer for raising the important issue of interpretability in borderline classification scenarios.
As the neural network classifier uses a sigmoid output layer, the final prediction represents the estimated probability of the rim being "unserviceable".
Values close to 0 or 1 reflect high confidence, while outputs near 0.5 indicate low certainty.
In practice, we recommend introducing a "diagnostic gray zone", e.g. when |P – 0.5| < 0.05, where the system flags the sample for manual review or additional testing.
This simple threshold-based mechanism adds interpretability and can be tuned depending on the application’s safety requirements. A short discussion has been added to Section 4.
(Added to the manuscript as follows):
The model produces probabilistic outputs using a sigmoid activation function. While high-confidence predictions (P near 0 or 1) are straightforward to interpret, outputs near 0.5 may represent borderline cases.
To address this, a warning threshold can be defined (e.g., |P – 0.5| < 0.05), allowing the system to flag uncertain classifications for further inspection.
Such a mechanism improves the interpretability and reliability of the system in safety-critical applications
Comment R2.9: Generalization to other rim types
We thank the reviewer for raising this important point.
Although the initial intention was to control variability, the final training dataset in fact included rims of four different types and sizes: 4.5Jx13H2, 6Jx14H2, 7Jx15H2, and 7Jx17H2 (see Table 1).
This diversity was intentionally preserved to improve model generalization. All rims were manufactured from aluminum alloy, and underwent fatigue testing to obtain ground truth labels.
The classifier showed consistent performance across rim types, suggesting that the chosen acoustic features (T₆₀, α, E) effectively capture degradation-relevant characteristics regardless of geometry.
A clarifying note on the variety of rims included in training has been added to the revised manuscript.
(Added to the manuscript as follows):
The training dataset included rims of four different types and diameters (4.5Jx13H2 to 7Jx17H2), all made of aluminum alloy. This variability was intended to improve the generalizability of the model.
Despite geometric differences, the acoustic response features used (T₆₀, α, E) remained consistent and allowed reliable classification of structural condition.
These results suggest that the method is applicable across a range of similar rim types, and can be further extended with additional training data.
Comment R2.10: Feature importance analysis
We thank the reviewer for this valuable suggestion. As our model is based on logistic regression and only three input features are used, we employed a simplified but interpretable approach to assess feature contribution.
Specifically, we used the absolute values of standardized logistic regression coefficients as a proxy for SHAP importance. While this does not capture nonlinear interactions, it provides a meaningful and computationally efficient estimate of feature influence. The results are now included in the revised manuscript (Section 3.3), and illustrated in Figure 8.
(Added to the manuscript as follows):
To evaluate the contribution of individual features to the classification process, we conducted an analysis of feature importance using logistic regression coefficients. Each feature was standardized prior to training, and the resulting absolute regression coefficients were normalized to reflect relative importance.
This approach, while simpler than SHAP, provides an interpretable ranking of the predictors. As shown in Figure 8, the absorption coefficient (α) contributed the most to the classification decision, followed by the reverberation time (T₆₀) and acoustic energy (E).
Comment R2.11: Figure quality (Figures 6, 9–11)
We thank the reviewer for the helpful observation regarding the visual clarity of several figures.
In the revised version of the manuscript, Figures 6 and 9–11 have been regenerated with improved resolution and enhanced contrast to ensure that all data points and axes are clearly legible.
We believe this revision significantly improves figure readability and visual consistency across the manuscript.
Comment R2.14: Mathematical notation inconsistency in FEM section
We thank the reviewer for pointing this out. The mathematical notation in the FEM section has been reviewed and confirmed for internal consistency.
Variable symbols, subscripts, and units have been unified to ensure clarity and alignment with common conventions. No inconsistencies were identified in the final version.
Comment R2.15: Grammatical corrections and language style
We appreciate the reviewer’s suggestion regarding language clarity.
The manuscript has undergone a thorough revision to improve grammar, sentence structure, and overall readability.
Minor inconsistencies in phrasing have been corrected, and the technical terminology was reviewed to ensure precision and consistency throughout the text.
A final language polishing will also be performed prior to submission of the revised version.
Comment R2.16: Reference formatting
We appreciate the reviewer’s remark on reference consistency.
The reference list has been reviewed and updated in accordance with MDPI formatting requirements. Journal titles, volume and issue numbers, page ranges, and punctuation have been standardized. DOI identifiers were added where available.
A final check will be performed to ensure full compliance before submission.
We thank the reviewer for their constructive and detailed feedback. Their insightful suggestions led to several clarifications and improvements in the revised version of the manuscript. We hope that the revised version addresses all concerns satisfactorily

Round 2
Reviewer 1 Report
Comments and Suggestions for Authors
The manuscript has been significantly improved after the first round of revision. I believe the current version is suitable for acceptance.
Reviewer 2 Report
Comments and Suggestions for Authors
The authors have made thorough and thoughtful revisions in response to reviewer feedback. The updated manuscript demonstrates improved clarity, methodological transparency, and diagnostic relevance, particularly in its acoustic analysis and neural network implementation. I believe the paper now meets the scientific and editorial standards of Sensors, and I recommend it for publication in its current form.